# Distributed Constrained Optimal Consensus Under a Directed Graph

## Abstract

In this paper, the distributed constrained optimal consensus problem of multi-agent systems under a directed graph is investigated. We propose two projection-based distributed constrained optimal consensus algorithms: one addressing set constraints and the other tailored for general constraints. Only the relative state is exchanged among agents in these two algorithms. In the stability analysis of case with set constraints, we transform the distributed optimization problem into a constrained leaderless consensus problem by adopting a sliding mode approach. Building on this foundational transformation, we further develop a projection-based distributed constrained optimal consensus algorithm to address general constraints. It is shown that the proposed algorithm achieves an ergodic convergence rate of $O(\frac{1}{k})$ with respect to the first-order optimality residuals. Numerical simulations are conducted to validate the effectiveness of our theoretical results.

## 1 INTRODUCTION

In the past two decades, distributed optimization of multi-agent systems has drawn extensive attention. As a fundamental problem of distributed optimization, distributed optimal consensus refers to construct an algorithm for each agent such that the states of agents converge to the optimal solution of a global objective function using local information. Early distributed optimal consensus of multi-agent systems can be referred to (Shi et al., 2015) and (Kia et al., 2015). The distributed optimization algorithm has diverse practical applications, including cooperative transportation (Chen & Kai, 2018), energy management (Wang et al., 2024), distributed learning (Arjevani & Shamir, 2015). For a comprehensive overview of recent advances in distributed optimization, one can refer to (Yang et al., 2019; Shorinwa et al., 2024).

### 1.1 RELATED WORK

Many previous studies have developed distributed optimization algorithms under different types of constraints. In relation to the work presented in this paper, we categorize the current advancements into three groups based on the types of constraints as follows:

**Unconstrained optimization:** Distributed discrete time unconstrained optimization algorithms primarily include DGD (Nedic & Ozdaglar, 2009; Tu et al., 2022), EXTRA (Shi et al., 2015), DEXTRA (Xi & Khan, 2017), Exact-Diffusion (Yuan et al., 2018; Alghunaim, 2024), Gradient-Tracking (Pu & Nedić, 2021), among others. Compared with the vanishing step-sizes in DGD, EXTRA, Exact-Diffusion, and Gradient-Tracking uses constant step-size and has linear convergence speed $O(C^{-k})$. Be aware that due to the introduction of an integral term used to compensate gradient in equilibrium, both EXTRA and DEXTRA demonstrate second-order dynamics.

**Optimization with set constraints:** Set constraints are considered in (Nedic et al., 2010; Liu & Wang, 2015; Scaman et al., 2018; Mai & Abed, 2019; YU2, 2021). Distributed optimization problems with set constraints under an undirected graph are considered in (Nedic et al., 2010; Liu & Wang, 2015; Scaman et al., 2018). Early projection-based algorithm considering is presented in (Nedic et al., 2010). (Liu & Wang, 2015) proposes a distributed proportional-integral (PI) type projection-based gradient method. Conic constraint sets are considered in (Scaman et al., 2018). Different from the undirected graph case, general directed graph cases are considered in (Mai & Abed, 2019; YU2, 2021; Wan et al., 2020). Due to the second-order dynamics in EXTRA, DEX-

TRA, Exact-Diffusion, it is of challenge for incorporating projection algorithms in environments with set constraints. This is precisely why the existing algorithms with set constraints adopt vanishing step-sizes in (Mai & Abed, 2019; YU2, 2021; Wan et al., 2020). Moreover, vanishing step-sizes are used in distributed gradient-tracking algorithm due to different set constraints (Cheng et al., 2023). One drawback of utilizing diminishing step sizes is a slower convergence rate. Therefore, it is desired to design a projection based algorithm with constant step-sizes.

**Optimization with general constraints:** Except for the set constraints considered in (Nedic et al., 2010; Liu & Wang, 2015; Scaman et al., 2018; Mai & Abed, 2019; YU2, 2021; Wan et al., 2020), general constraints include equality constraints and nonlinear inequality constraints, which can be solved by primal-dual method (Gao, 2004). These nonlinear constraints bring more challenges to the distributed optimization algorithm design. The existing algorithms on distributed optimization algorithms addresses both the undirected graph scenario (Yang et al., 2016; Zhu et al., 2018) and the directed graph scenario (Khatana & Salapaka, 2023). Besides the difficulties of projection operation, the existence of nonlinear inequality constraints has introduced significant complexities into stability analysis, which is not considered in (Khatana & Salapaka, 2023).

## 1.2 CONTRIBUTION

From the above analysis, it becomes clear that existing distributed optimization algorithms have not yet tackled scenarios that simultaneously incorporate directed topology graphs, fixed step sizes, and general constraints. In this paper, we study the distributed constrained optimal consensus problem of multi-agent systems under a directed graph using only relative state. The main challenge lies in the fact that the nonlinear projection operator and nonlinear constraints hinder the stability analysis. To address this challenge, we introduce an original analysis framework based on the sliding mode technique, which transforms the distributed constrained optimal consensus problem into a constrained leaderless consensus problem. Compared with existing results (Nedic et al., 2010; Liu & Wang, 2015; Scaman et al., 2018; Mai & Abed, 2019; YU2, 2021; Wan et al., 2020; Gao, 2004; Yang et al., 2016; Zhu et al., 2018; Khatana & Salapaka, 2023), the proposed algorithm makes the following contributions:

**Case with set constraints:** we develop a distributed algorithm using only the interaction of relative state under a balanced directed graph (Theorem 3.6). By using a sliding mode approach, we transform the distributed constrained optimal consensus problem into a constrained leaderless consensus problem. Within this transformed framework, we design a novel Lyapunov function and conduct a comprehensive stability analysis to illustrate the effectiveness of our approach. Different from undirected graphs considered in (Nedic et al., 2010; Liu & Wang, 2015; Scaman et al., 2018), we consider the directed graph case. Compared with the methods with vanishing step-sizes (Mai & Abed, 2019; YU2, 2021), our algorithm utilize the constant step-sizes which can achieve faster convergence speed. By omitting local set constraints, our algorithm can be conducted under a directed graph case, which is more general than EXTRA (Shi et al., 2015).

**Case with general constraints:** A projection-based algorithm is proposed for the case with general constraints (Theorem 4.4). Different from the undirected graph case, our proposed algorithm (Yang et al., 2016; Zhu et al., 2018) can be deployed under a directed graph. Compared with (Khatana & Salapaka, 2023), we only communicate the relative state information. Unlike the single system optimization algorithm in (Gao, 2004), we propose a novel algorithm to tackle general constraints.

## 2 BACKGROUND AND PROBLEM STATEMENT

### 2.1 GRAPH THEORY

We use a directed graph to describe the network topology among $N$ agents. Let $\mathcal{G}_N \triangleq (\mathcal{V}_N, \mathcal{E}_N)$ be a directed graph with the node set $\mathcal{V}_N \triangleq \{v_1, ..., v_N\}$ and the edge set $\mathcal{E}_N \subseteq \mathcal{V}_N \times \mathcal{V}_N$. An edge $(v_i, v_j) \in \mathcal{E}_N$ denotes that agent $v_j$ can obtain information from agent $v_i$, but not vice versa. Here, node $v_i$ is the parent node while node $v_j$ is the child node. A directed path from node $v_i$ to node $v_j$ is a sequence of edges of the form $(v_i, v_{i,2}), (v_{i,2}, v_{i,3}), \ldots, (v_{i,k}, v_j)$, in a directed graph. A directed tree is a directed graph, where every node has exactly one parent except for one node,

called the root, and the root has directed paths to every other node. A directed spanning tree of a directed graph is a direct tree that contains all nodes of the directed graph. A directed graph contains a directed spanning tree if there exists a directed spanning tree as a subset of the directed graph. A directed graph is balanced graph iff each node has the same in-degree and out-degree. For balanced graphs, $\mathcal{L}_N^T \mathbf{1}_N = \mathbf{0}_N$, and $\mathcal{L}_N + \mathcal{L}_N^T$ is a symmetric Laplacian matrix of some undirected graph.

The adjacency matrix $\mathcal{A}_N \overset{\triangle}{=} [a_{ij}] \in \mathbb{R}^{N \times N}$ of a directed graph $(\mathcal{V}_N, \mathcal{E}_N)$ is defined such that $a_{ij} > 0$ if $(v_j, v_i) \in \mathcal{E}_N$, and $a_{ij} = 0$ if $(v_j, v_i) \notin \mathcal{E}_N$. In this paper, self-edges are not allowed, *i.e.*, $a_{ii} = 0$. The in-degree of node $i$ is defined as follows: $\deg_{in}(v_i) = \sum_{j=1}^{N} a_{ij}, i = 1, ..., N$. Then, $\mathcal{D}_N = \text{diag} \{\deg_{in}(v_1), \cdots, \deg_{in}(v_N)\}$ is called the degree matrix of $\mathcal{G}_N$. The (non-symmetric) Laplacian matrix of $\mathcal{G}_N$ is defined as $\mathcal{L}_N = \mathcal{D}_N - \mathcal{A}_N$, with $l_{ii} = \sum_{j=1,j\neq i}^{n} a_{ij}$ and $l_{ij} = -a_{ij}$, $i \neq j$.

**Assumption 2.1** *The directed graph $\mathcal{G}$ is strongly connected and balanced.*

We define a vector $r_N = (\frac{1}{N}, \frac{1}{N}, \ldots, \frac{1}{N}) \in \mathbb{R}^{1 \times N}$ and a matrix $R_N \in \mathbb{R}^{(N-1) \times N}$

$$
R_N = \begin{pmatrix}
-1 + (N-1)v & 1-v & -v & \cdots & -v \\
-1 + (N-1)v & -v & 1-v & \ddots & \vdots \\
\vdots & \vdots & \ddots & \ddots & -v \\
-1 + (N-1)v & -v & \cdots & -v & 1-v
\end{pmatrix}, \tag{1}
$$

where $v = \frac{N - \sqrt{N}}{N(N-1)}$. Then, we have that $U = \begin{pmatrix} r_N \\ R_N \end{pmatrix} \in \mathbb{R}^{N \times N}$ is a unitary matrix, with $R_N \mathbf{1}_N = 0, R_N R_N^T = \mathbf{I}_{N-1}$, and $R_N^T R_N = I_N - \frac{1}{N} \mathbf{1}_N \mathbf{1}_N^T$. By observing that $\tilde{x} \overset{\triangle}{=} R_N x$ is equal to zero if and only if $x = \alpha \mathbf{1}_N$ for $\alpha \in \mathbb{R}$, which implies that the null space of $R_N$ is $\mathbf{1}_N$. $\|\tilde{x}\|$ can be viewed as a measure of synchrony. When $\|\tilde{x}\| = 0$, the agents' states achieve consensus.

## 2.2 MAIN LEMMAS

**Lemma 2.1** *(Scardovi et al., 2010) For a Laplacian matrix $\mathcal{L}_N$ of graph $\mathcal{G}_N$ with a spinning tree, $-R_N \mathcal{L}_N R_N$ is a Hurwitz matrix. Moreover, if $\mathcal{G}_N$ is undirected, we have $R_N \mathcal{L}_N R_N$ is negative-definite and $x^T \mathcal{L}_N x = x^T U^T U \mathcal{L}_N U^T U x = x^T R_N^T R_N \mathcal{L}_N R_N^T R_N x = \tilde{x}^T R_N \mathcal{L}_N R_N^T \tilde{x}$.*

**Theorem 2.2** *(Theorem 3.14, (Rudin et al., 1964)) If a sequence of real numbers is decreasing and bounded below, then its infimum is the limit.*

## 2.3 CONVEX FUNCTION ANALYSIS

To facilitate the stability analysis, we have the following assumptions about the local objective function. Let $f \in C^{1,1}(\mathbb{R}^p)$, *i.e.*, $f : \mathbb{R}^p \to \mathbb{R}$ is a continuously differentiable function and its derivative $\nabla f(x)$ is locally Lipschitz continuous on $\mathbb{R}^p$. $\nabla f(x)$ is locally Lipschitz continuous if for any compact set $U \subset \mathbb{R}^p$, *i.e.*, there always exists a positive constant $M_l \geq 0$ such that $\|\nabla f(x) - \nabla f(y)\| \leq M_l \|x - y\|$ for $y, z \in U$. $f(x)$ is $m$-strongly convex iff $(\nabla f(y) - \nabla f(x))^T (y - x) \geq m\|y - x\|^2, \forall x, y \in \mathbb{R}^p$, where $m > 0$ is the strong convexity constant. $f(x)$ has $M_f$-Lipschitz gradient iff $\|\nabla f(y) - \nabla f(x)\|^2 \leq M_f \|y - x\|^2, \forall x, y \in \mathbb{R}^p$, where $M_f > 0$ is Lipschitz constant.

For a $C^{1,1}$ function $f(x)$, $\nabla f(x)$ is differentiable almost everywhere so that its generalized derivative in Clarke's sense can be defined everywhere. The definition of the general Hessian matrix is given as follows.

**Definition 2.3** *(Hiriart-Urruty et al., 1984) The generalized Hessian matrix of $f$ at $x_0$, denoted by $\partial^2 f(x_0)$, is the set of matrices defined as the convex hull of the set*

$$
\overline{co}\{M \mid \exists x_i \to x_0 \text{ with } f \text{ twice differentiable at } x_i \text{ and } \nabla^2 f(x_i) \to M\}.
$$

*where $\overline{co}$ denotes the convex closure. From the above definition, $\partial^2 f(x_0)$ is a nonempty compact convex set.*

## 2.4 CONVEX SET ANALYSIS

A set $\Omega \in \mathbb{R}^n$ is called convex if $\lambda x + (1 - \lambda)y \in \Omega$, $\forall x, y \in \Omega$, and $\lambda \in (0, 1)$. Given a closed convex set $\Omega \subseteq \mathbb{R}^p$, the projection $P_\Omega(x) : \mathbb{R}^p \to \Omega$ is defined as $P_\Omega(x) = \text{argmin}_{y \in \Omega} \|x - y\|$. For a convex set $\Omega$, the projection variable satisfies the Lipschitz continuity $\|P_\Omega(x) - P_\Omega(y)\| \leq \|x - y\|$.

**Lemma 2.2** *(Khalil, 2002) Given a closed convex set $\Omega \subseteq \mathbb{R}^n$, $\mu = P_\Omega(v)$ if and only if $\mu \in \Omega$ and $(\mu - v)^T(y - \mu) \geq 0$ for any $y \in \Omega$.*

Clearly, by replacing $\mu$ with $P_\Omega(y)$ in the inequality of Lemma, it follows that

$$(P_\Omega(y) - y)^T(z - P_\Omega(y)) \geq 0, \ \forall z \in \Omega. \tag{2}$$

**Definition 2.4** *The normal cone at point $x$ with respect to the convex set $\Omega$ is defined as:*

$$N_\Omega(x) = \{v \in \mathbb{R}^p \mid \langle v, y - x \rangle \leq 0 \text{ for all } y \in \Omega\}.$$

Normal cone is always convex and is closed under positive scalar multiplication, *i.e.*, for any $x, y \in N_\Omega(x)$, $t_1 x + t_2 y \in N_\Omega(x)$, $\forall t_1, t_2 > 0$.

**Lemma 2.3** *(Kinderlehrer & Stampacchia, 1980) For a convex set $\Omega \in \mathbb{R}^p$, we can define the following function $V_{x,y} = \frac{1}{2}\|x - P_\Omega(y)\|^2 - \frac{1}{2}\|x - P_\Omega(x)\|^2$ which satisfies*

*(1) $V_{x,y} \geq \frac{1}{2}\|P_\Omega(x) - P_\Omega(y)\|^2$;*

*(2) $V_{x,y}$ is continuously differentiable and convex with respect to $x \in \mathbb{R}^p$, and $\nabla_x V_{x,y} = P_\Omega(x) - P_\Omega(y)$.*

# 3 DISTRIBUTED OPTIMAL CONSENSUS ALGORITHM WITH SET CONSTRAINTS

In this section, we aim to design a distributed projection-based optimal consensus algorithm for multiple first-order integrators using relative state information, without the need to communicate virtual variables with neighbors. Different from the unconstrained case, the main challenge in the algorithm design lies in the nonlinearity of the projection method.

## 3.1 PROBLEM STATEMENT

Each agent is assigned a local objective function $f_i : \mathbb{R}^p \to \mathbb{R}$ and a local set constraint $\Omega_i$ by a nonempty closed convex set of $\mathbb{R}^p$, which are only known by itself. The global objective function is defined by $\sum_{i=1}^n f_i(q)$. Our aim is to design a distributed optimal consensus algorithm for such that the state of each agent converges to the optimal solution of following problem

$$\min \sum_{i=1}^n f_i(x),$$
$$s.t. \ x \in \bigcap_{i=1,\ldots,n} \Omega_i, \tag{3}$$

using only relative state information, the local objective function information, and the local set constraint.

Next, the following assumptions and definitions provides the optimality condition to the constrained optimization problem equation (3).

**Assumption 3.1** *(Existence) Let $\Omega$ be the intersection of all local closed convex sets. $\Omega$ is a non-empty set, i.e., $\Omega = \bigcap_{i=1}^n \Omega_i \neq \emptyset$.*

From Assumption 3.1, if there exists $x \in \Omega$ and $y \in N_\Omega(x)$, then $y \in N_{\Omega_i}(x) \subseteq N_\Omega(x)$ due to $\Omega \subseteq \Omega_i$, for all $i = 1, \ldots, n$.

**Assumption 3.2** *The local gradient of objective function $\nabla f_i(x)$, and the gradient of nonlinear inequality constraint function $\nabla g_{ik}(x)$ is $M_i$-Lipschitz satisfying $H_i(x_i) \preceq M_i \mathbf{I}_p$, $H_{gi}(x_i) \preceq M_{gi} \mathbf{I}_p$, $\forall H_i(x_i) \in \partial^2 f_i(x_i)$, $\forall H_{gi}(x_i) \in \partial^2 g_i(x_i)$, $\forall x_i \in \mathbb{R}^p$.*

From Assumption 3.2, $f_i(y) \leq f_i(x) + \nabla f_i^T(x)(y - x) + \frac{M_i}{2}\|y - x\|^2$.

**Assumption 3.3** *The local objective function $f_i(x)$ is differentiable and $m_i$-strongly convex which means that any matrix in generalized Hessian matrix $H_i(x_i) \succeq m_i \mathbf{I}_p$, $\forall H_i(x_i) \in \partial^2 f_i(x_i)$, $\forall x_i \in \mathbb{R}^p$.*

From Assumption 3.3, we also have $f_i(y) \geq f_i(x) + \nabla f_i^T(x)(y - x) + \frac{m_i}{2}\|y - x\|^2$.

**Definition 3.4** *$x^*$ is the optimal solution to the constrained optimization problem equation (3), if and only if $x^* - P_\Omega(x^* - \sum_{i=1}^n \nabla f_i(x^*)) = \mathbf{0}_p$. Or equivalently, $\sum_{i=1}^n \nabla f_i(x^*) \in N_\Omega(x^*)$.*

**Definition 3.5** *The distributed constrained optimal consensus problem is solved if for any initial condition $x_i(0) \in \mathbb{R}^p$, all the agents can converge to the global optimal solution of equation (3), i.e.,*

$$\lim_{t \to \infty} x_i(t) = x^*. \tag{4}$$

### 3.2 ALGORITHM DESIGN

We introduce the following consensus algorithm

$$x_i(k+1) = x_i(k) - T(x_i(k) - P_{\Omega_i}(x_i(k) - \alpha_i(\sum_{j=1}^n a_{ij}(x_i(k) - x_j(k)) + \nabla f_i(x_i(k))$$

$$+ w_i(k))))$$

$$w_i(k+1) = w_i(k) - T\sum_{j=1}^n a_{ij}(x_i(k) - x_j(k)). \tag{5}$$

*Equilibrium analysis:* Let the equilibrium of $(x_i(k), w_i(k))$ be $(x_i^*, w_i^*)$. Then, we have

$$\mathbf{0}_p = -T(x_i^*(k) - P_{\Omega_i}(x_i^*(k) - \alpha_i(\sum_{j=1}^n a_{ij}(x_i^* - x_j^*) + \nabla f_i(x_i^*) + w_i^*)),$$

$$\mathbf{0}_p = T\sum_{j=1}^n a_{ij}(x_i^* - x_j^*). \tag{6}$$

From equation (6), we have $x_i^* = x_j^*$ and $-\alpha_i(\nabla f_i(x^*) + w_i^*) \in N_{\Omega_i} \subseteq N_\Omega$, $\forall i, j = 1, \ldots, n$. Exploiting the convexity property of the normal cone at $x_i^*$, we can conclude that $\sum_{i=1}^n \alpha_i^{-1}[-\alpha_i(\nabla f_i(x_i^*) + w_i^*)] = -\sum_{i=1}^n \nabla f_i(x_i^*) \in N_{\Omega_i} \subseteq N_\Omega$. Therefore, $\sum_{i=1}^n \nabla f_i(x_i^*) = -\sum_{i=1}^n w_i^* = \mathbf{0}_p$ and $x_i^* = x^*$.

*Stability analysis:* Define the following auxiliary variables

$$s_i(k) = x_i(k) - \alpha_i(\sum_{j=1}^n a_{ij}(x_i(k) - x_j(k)) + \nabla f_i(x_i(k)) + w_i(k)),$$

$$\mu_i(k) = P_{\Omega_i}(s_i(k)).$$

If $\Omega_i = \mathbb{R}^p$, $\mu_i$ will satisfy $\mu_i(k) = x_i(k) + \frac{1}{T}(x_i(k+1) - x_i(k))$ and can be viewed as a sliding variable. Therefore, we can regard $\mu_i(k)$ as a projected sliding variable. By introducing $\mu_i(k)$, the dynamics of $x_i$ in equation (5) can be written as

$$x_i(k+1) = x_i(k) - T(x_i(k) - \mu_i(k)). \tag{7}$$

equation (7) can be understood as a first-order dynamics of $x_i$ tracking $\mu_i$. Given that $\mu_i$ is the projection of $y_i$ onto $\Omega_i$, taking the difference of $y_i$ yields

$$s_i(k+1) - s_i(k) = T(\mu_i(k) - x_i(k)) - T\alpha_i \sum_{j=1}^n a_{ij}(\mu_i(k) - \mu_j(k)) - \alpha_i(h_i(k+1) - h_i(k)).$$

(8)

Define $\tilde{\mu}_i = \mu_i - x^*$, $\tilde{x}_i = x_i - x^*$. Then, the error dynamics can be written as

$$\tilde{x}_i(k+1) = \tilde{x}_i(k) - T(\tilde{x}_i(k) - \tilde{\mu}_i(k))$$

$$s_i(k+1) = s_i(k) + T(\tilde{\mu}_i(k) - \tilde{x}_i(k)) - T\alpha_i \sum_{j=1}^n a_{ij}(\tilde{\mu}_i(k) - \tilde{\mu}_j(k)) - \alpha_i(h_i(k+1) - h_i(k)).$$

(9)

Note that equation (9) can be viewed as a constrained leaderless consensus problem, where $\mu_i$ satisfies the local constraint $\mu_i \in \Omega_i$. Define the stack vectors $\tilde{\mu} = (\tilde{\mu}_1^T, \ldots, \tilde{\mu}_n^T)^T$, $s = (s_1^T, \ldots, s_n^T)^T$, $\tilde{x} = (\tilde{x}_1^T, \ldots, \tilde{x}_n^T)^T$. The vector form of equation (9) can be written as

$$\tilde{x}(k+1) = \tilde{x}(k) - T(\tilde{x}(k) - \tilde{\mu}(k))$$
$$s(k+1) - s(k) = T(\tilde{\mu}(k) - \tilde{x}(k)) - T\alpha\mathcal{L}\tilde{\mu}(k) - \alpha(h(k+1) - h(k)). \qquad (10)$$

**Theorem 3.6** *Under Assumptions 2.1, 3.1, 3.2,and 3.3, the agents by using equation (5) will eventually converge to the optimal solution of problem equation (3), i.e., $\lim_{t\to\infty}[x_i(t) - \mu^*] = \mathbf{0}_p$, $i = 1, \ldots, n$.*

*Proof:* The detail of proof is given in Appendix A.1. ∎

## 4 DISTRIBUTED OPTIMAL CONSENSUS ALGORITHM WITH GENERAL CONSTRAINTS

In this section, we aim to design a distributed projection-based optimal consensus algorithm using relative state information, without the need to communicate virtual variables between neighbors. Different from the case with only set constraints, the introduction of additional equality and nonlinear inequality constraints bring difficulties to the algorithm design.

### 4.1 PROBLEM STATEMENT

Our aim is to design a distributed optimal consensus algorithm such that the state of each agent achieves the optimal solution of following problem

$$\min \sum_{i=1}^n f_i(x)$$

$$s.t. \ x \in \bigcap_{i=1,\ldots,n} \Omega_i, \ g_{ik}(x) \le 0,$$

$$A_i x = b_i, \ i = 1, \ldots, n, \ k = 1, \ldots, p_{g_i}. \qquad (11)$$

using relative state measurements, the local objective function information, and the local general constraints. Here, $g_{ik}(x)$ is first-order differentiable convex function $A_i \in \mathbb{R}^{p_{b_i} \times p}$. Next, the following assumptions and definitions provides the optimality condition to the constrained optimization problem equation (11)

**Assumption 4.1** *(Slater's condition) Let $\Omega$ be the intersection of all the local closed convex sets. $\Omega$ is a non-empty set, i.e., $\Omega = \bigcap_{i=1}^n \Omega_i \ne \emptyset$. Furthermore, there exist $x \in \mathbb{R}^p$ such that*

$$x \in \mathrm{relint}(\Omega), \ g_{ik}(x) < 0, \ A_i x = b_i, \ i = 1, \ldots, n, \ k = 1, \ldots, p_{g_i}.$$

where $\mathrm{relint}(\cdot)$ denotes the relative interior of a set. From Assumption 4.1, if there exists $x \in \Omega$ and $y \in N_\Omega(x)$, then $y \in N_{\Omega_i}(x) \subseteq N_\Omega(x)$ due to $\Omega \subseteq \Omega_i$.

**Assumption 4.2** *The gradient of nonlinear inequality constraint function $\nabla g_{ik}(x)$ is $M_{gi}$-Lipschitz satisfying $H_{gi}(x_i) \preceq M_{gi}\mathbf{I}_p$, $\forall H_{gi}(x_i) \in \partial^2 g_i(x_i)$, $\forall x_i \in \mathbb{R}^p$.*

**Lemma 4.1** *(Yang et al., 2016) (KarushKuhnTucker (KKT) conditions) Under Assumptions 4.1 and 3.3, $x^*$ is the optimal solution to the constrained optimization problem equation (11), if and only if there exist $\gamma_{ik}^* \in \mathbb{R}^+$, $\nu_i^* \in \mathbb{R}^{p_{b_i}}$, $x^* \in \Omega$ such that*

$$-\sum_{i=1}^{n} \left[ \nabla f_i(x^*) - \sum_{k=1}^{p_{g_i}} \gamma_{ik}^* \nabla g_{ik}(x^*) - A_i^T \nu_i^* \right] \in N_\Omega(x^*),$$

$$g_{ik}(x^*) \leq 0, \ \gamma_{ik}^* \geq 0, \ \gamma_{ik}^* g_{ik}(x^*) = 0,$$

$$A_i x^* = b_i, \forall i = 1, \ldots, n, \ k = 1, \ldots, p_{g_{ik}}.$$

## 4.2 Distributed Optimal Consensus Algorithm Considering General Constraints

We introduce the following auxiliary variables to address specific types of constraints within the optimization problem: $w_i$ corresponds to the consensus constraint, $\gamma_{ik} \in \mathbb{R}^+$ corresponds to the inequality constraints, and $\nu_i \in \mathbb{R}^{p_{b_i}}$ corresponds to the equality constraint, $i = 1, \ldots, n$, $k = 1, \ldots, p_{g_i}$. Motivated by (11) in (Zhu et al., 2018), we introduce the following projection-based algorithm

$$x_i(k+1) = x_i(k) - T\Big[x_i(k) + P_{\Omega_i}(x_i(k) - \alpha_i(\sum_{j=1}^{n} a_{ij}(x_i(k) - x_j(k)) + \nabla f_i(x_i(k)) + w_i(k)$$

$$+ \sum_{k=1}^{p_{g_i}} \gamma_{ik}(k)\nabla g_{ik}(x_i(k)) + A_i^T \nu_i(k) + \alpha_{\nu i} A_i^T (A_i x_i(k) - b_i)))\Big], \tag{12}$$

and the differential dynamics of auxiliary variables is described by the following algorithm

$$w_i(k+1) = w_i(k) + T\sum_{j=1}^{n} a_{ij}(x_i(k) - x_j(k)),$$

$$\gamma_{ik}(k+1) = (1 - \alpha_{\gamma i}T)\gamma_{ik}(k) + T\alpha_{\gamma i}\Big\{\gamma_{ik}(k) + g_{ik}(x_i(k)) - \nabla g_{ik}^T(x_i(k))\big[x_i(k)$$

$$- P_{\Omega_i}(x_i(k) - \alpha_i(\sum_{j=1}^{n} a_{ij}(x_i(k) - x_j(k)) + \nabla f_i(x_i(k)) + w_i(k)$$

$$+ \sum_{k=1}^{p_{g_i}} \gamma_{ik}(k)\nabla g_{ik}(x_i(k)) + A_i^T \nu_i(k) + \alpha_{\nu i} A_i^T (A_i x_i(k) - b_i)))\big]\Big\}^+,$$

$$\nu_i(k+1) = \nu_i(k) + T\alpha_{\nu i}(A_i x_i(k) - b_i). \tag{13}$$

with $w_i(0) = \mathbf{0}_p$, $\gamma_{ik}(0) \geq 0$, $\nu_i(0) \in \mathbb{R}^{p_{b_i}}$, and $\alpha_{\gamma i}, \alpha_{\nu i} \in \mathbb{R}$ being positive gains used to adjust convergence speed.

*Equilibrium analysis:* Let the equilibrium of $(x_i, w_i, \gamma_{ik}, \nu_i)$ be $(x_i^*, w_i^*, \gamma_{ik}^*, \nu_i^*)$. Then, we have

$$
\mathbf{0}_p = -x_i^* + P_{\Omega_i}(x_i^* - \alpha_i(\sum_{j=1}^{n} a_{ij}(x_i^* - x_j^*) + \nabla f_i(x_i^*) + w_i^* + \sum_{k=1}^{p_{g_i}} \gamma_{ik}^* \nabla g_{ik}(x_i^*) + A_i^T \nu_i^*
$$

$$
+ \alpha_{\nu i} A_i^T (A_i x_i^* - b_i))),
$$

$$
\mathbf{0}_p = \sum_{j=1}^{n} a_{ij}(x_i^* - x_j^*),
$$

$$
0 = -\alpha_{\gamma i} \gamma_{ik}^* + \alpha_{\gamma i}\Big\{\gamma_{ik}^* + g_{ik}(x_i^*) - \nabla g_{ik}^T(x_i^*)\big[x_i^* - P_{\Omega_i}(x_i^* - \alpha_i(\sum_{j=1}^{n} a_{ij}(x_i^* - x_j^*)
$$

$$
+ \nabla f_i(x_i^*) + w_i^* + \sum_{k=1}^{p_{g_i}} \gamma_{ik}^* \nabla g_{ik}(x_i^*) + A_i^T \nu_i^* + \alpha_{\nu i} A_i^T (A_i x_i^* - b_i)))\big]\Big\}^{+},
$$

$$
\mathbf{0}_{p_{b_i}} = \alpha_{\nu i}(A_i x_i^* - b_i). \tag{14}
$$

From equation (14), we can conclude that $x_i^* \in \Omega_i$, $x_i^* = x_j^*$, $\gamma_{ik}^* = (\gamma_{ik}^* + g_{ik}(x_i^*))^+$, $A_i x_i^* - b_i = \mathbf{0}_{p_{b_i}}$, $\forall i, j = 1, \ldots, n$, $k = 1, \ldots, p_{g_i}$. Therefore, we obtain that $\gamma_{ik}^* \geq 0$, $g_{ik}(x_i^*) \leq 0$, and $\gamma_{ik}^* g_{ik}(x_i^*) = 0$. From $x_i^* \in \Omega_i$ and $x_i^* = x_j^*$, we find $x_i^* \in \bigcap_{i=1,\ldots,n} \Omega_i$. Then we have $-\alpha_i(\nabla f_i(x_i^*) + w_i^* + \sum_{k=1}^{p_{g_i}} \gamma_{ik}^* \nabla g_{ik}(x_i^*) + A_i^T \nu_i^*) \in N_{\Omega_i} \subseteq N_{\Omega}(x_i^*)$. Utilizing the convexity property of the normal cone, the equation $\sum_{i=1}^{n} \alpha_{xi}^{-1}[-\alpha_i(\nabla f_i(x_i^*) + w_i^* + \sum_{k=1}^{p_{g_i}} \gamma_{ik}^* \nabla g_{ik}(x_i^*) + A_i^T \nu_i^*)] = -\sum_{i=1}^{n}[\nabla f_i(x_i^*) + \sum_{k=1}^{p_{g_i}} \gamma_{ik}^* \nabla g_{ik}(x_i^*) + A_i^T \nu_i^*] \in N_{\Omega}(x_i^*)$ holds. According to Lemma 4.1, $x_i^*$ is the optimal solution $x^*$ of problem equation (11).

*Stability Analysis:* In this part, we have the following assumption

**Assumption 4.3** *$\gamma_{ik}(k)$ has upper bounds $\bar{\gamma}_{ik}$, the norm of $\nabla g_{ik}(x_i)$ has upper bound $\bar{g}_{ik}$.*

Assumption 4.3 is required because these two terms need to be bounded to determine the control gains. The determination of the upper bounds for $\gamma_{ik}(k)$ and $\nabla g_{ik}(x_i)$ requires continuous experimentation and testing, involving a process of trial and error to make progress.

Then, we introduce the following auxiliary variables

$$
y_{\mu i}(k) = x_i(k) - \alpha_i(\sum_{j=1}^{n} a_{ij}(x_i(k) - x_j(k)) + \nabla f_i(x_i(k)) + w_i(k)
$$

$$
+ \sum_{k=1}^{p_{g_i}} \gamma_{ik}(k) \nabla g_{ik}(x_i(k)) + A_i^T \nu_i(k) + \alpha_{\nu i} A_i^T (A_i x_i(k) - b_i)),
$$

$$
\mu_i(k) = P_{\Omega_i}(y_{\mu i}(k)).
$$

By introducing $\mu_i(k)$, the dynamics of $x_i(k)$ in equation (12) can be written as

$$
x_i(k+1) = x_i(k) - T(x_i(k) - \mu_i(k)), \tag{15}
$$

and the dynamics of $\gamma_{ik}$ in equation (13) can be written as

$$
\gamma_{ik}(k+1) = (1 - \alpha_{\gamma i} T)\gamma_{ik}(k) + T\alpha_{\gamma i}\big[\gamma_{ik}(k) + g_{ik}(x_i(k)) - \nabla g_{ik}^T(x_i(k))(\tilde{x}_i(k) - \tilde{\mu}_i(k))\big]^+. \tag{16}
$$

Since $\mu_i$ is the projection of $y_{\mu i}$ onto $\Omega_i$, we take the difference of $y_{\mu i}$ and obtain

$$
y_{\mu i}(k+1) \in y_{\mu i}(k) + T(\mu_i(k) - x_i(k)) - \alpha_i\Big[T\sum_{j=1}^{n} a_{ij}(\mu_i(k) - \mu_j(k)) + \nabla f_i(x_i(k+1))
$$

$$
- \nabla f_i(x_i(k)) + \sum_{k=1}^{p_{g_i}} \alpha_{\gamma i} \gamma_{ik}(k+1) \nabla g_{ik}(x_i(k+1)) - \sum_{k=1}^{p_{g_i}} \alpha_{\gamma i} \gamma_{ik}(k) \nabla g_{ik}(x_i(k))
$$

$$
+ T\alpha_{\nu i} A_i^T (A_i \mu_i(k) - b_i)\Big]. \tag{17}
$$

Note that for any $a \in \mathbb{R}^p$, $\exists \bar{x}_i(k) \in (x_i(k), x_i(k+1))$, $\exists \bar{\gamma}_{ik}(k) \in (\gamma_{ik}(k), \gamma_{ik}(k+1))$ such that

$$a^T[\gamma_{ik}(k+1)\nabla g_{ik}(x_i(k+1)) - \gamma_{ik}(k)\nabla g_{ik}(x_i(k))]$$

$$=a^T\bar{\gamma}_{ik}(k)\partial^2 g_{ik}(\bar{x}_i(k))(\tilde{x}_i(k+1) - \tilde{x}_i(k)) + a^T(\gamma_{ik}(k+1) - \gamma_{ik}(k))\nabla g_{ik}(\bar{x}_i(k))$$

$$=Ta^T\bar{\gamma}_{ik}(k)\partial^2 g_{ik}(\bar{x}_i(k))(\tilde{\mu}_i(k) - \tilde{x}_i(k)) + Ta^T\Big\{ -\alpha_{\gamma i}\gamma_{ik}(k) + \alpha_{\gamma i}\big[\gamma_{ik}(k) + g_{ik}(x_i(k))$$

$$- \nabla g_{ik}^T(x_i(k))(\tilde{x}_i(k) - \tilde{\mu}_i(k))\big]^+\Big\}\nabla g_{ik}(\bar{x}_i(k)). \tag{18}$$

where $\bar{x}_i(k)$ is weighted average of $x_i(k)$ and $x_i(k-1)$, $\bar{\gamma}_{ik}(k)$ is weighted average of $\gamma_{ik}(k)$ and $\gamma_{ik}(k+1)$.

Next, we begin to construct the error dynamics for equation (7) and equation (17). Let $\bar{\mu}$ be an interior point of feasible region according to Assumption 4.1, *i.e.*, $\bar{\mu} \in \text{relint}(\Omega)$, $g_{ik}(\bar{\mu}) < 0$, $A_i\bar{\mu} = b_i$. Define the errors as $\tilde{\mu}_i(k) = \mu_i(k) - \bar{\mu}$, $\tilde{x}_i(k) = x_i(k) - \bar{\mu}$. Note that we use $\bar{\mu}$ rather than $x^*$ in the errors to facilitate the subsequent Lyapunov function design. Since we transform the distributed constrained optimization problem equation (12) into a constrained leaderless consensus problem equation (7) and equation (17) in the stability analysis, it has no influence to apply the same displacement $\bar{\mu} - x^*$ to every $\tilde{\mu}_i(k)$ and $\tilde{x}_i(k)$. Then, the error dynamics can be written as

$$\tilde{x}_i(k+1) = \tilde{x}_i(k) - T(\tilde{x}_i(k) - \tilde{\mu}_i(k)),$$

$$y_{\mu i}(k+1) \in y_{\mu i}(k) + T(\tilde{\mu}_i(k) - \tilde{x}_i(k)) - \alpha_i\Big[T\sum_{j=1}^{n} a_{ij}(\tilde{\mu}_i(k) - \tilde{\mu}_j(k)) + \nabla f_i(x_i(k+1))$$

$$- \nabla f_i(x_i(k)) + \sum_{k=1}^{p_{g_i}} \alpha_{\gamma i}\gamma_{ik}(k+1)\nabla g_{ik}(x_i(k+1)) - \sum_{k=1}^{p_{g_i}} \alpha_{\gamma i}\gamma_{ik}(k)\nabla g_{ik}(x_i(k))$$

$$+ T\alpha_{\nu i}A_i^T A_i\tilde{\mu}_i(k)\Big],$$

$$\gamma_{ik}(k+1) = (1 - \alpha_{\gamma i}T)\gamma_{ik}(k) + T\alpha_{\gamma i}\big[\gamma_{ik}(k) + g_{ik}(x_i(k)) - \nabla g_{ik}^T(x_i(k))(\tilde{x}_i(k) - \tilde{\mu}_i(k))\big]^+. \tag{19}$$

**Theorem 4.4** *Under Assumptions 2.1, 3.3, 3.2, 4.1, 4.2, and 4.3, using equation (12) and equation (13), the agents will eventually achieve the optimal solution of problem equation (11), i.e.,* $\lim_{t\to\infty}[x_i(t) - x^*] = \mathbf{0}_p$, $i = 1, \ldots, n$.

*Proof:* The detail of proof is given in Appendix A.2

∎

## 5 NUMERICAL RESULTS

We use numerical simulations on a group of 20 agents to verify the effectiveness of the proposed algorithm. We conduct two algorithm simulations, one group has set constraints and another group has general constraints.

We conduct out our algorithm under different balanced graphs, directed circle graph, random directed graph with 180 edges, and complete graph to evaluate the algorithm convergence speed under various topology. The local objective function is defined in the form of $C^{1,1}$ quadratic function $f_i(x_i) = \int_0^{x_i - x_i^*} (x_i - x_i^*)^T A_i \mathrm{d}x$, $i = 1, \ldots, 20$, which is the strongly convex case presented in Example 1 of (Hiriart-Urruty et al., 1984). The set constraints are selected as $\Omega_i = \{x_2| x_{i,2} \leq 0.6\}$, the equality constraints are selected as $x_{i,1} = 0.5$, and the inequality constraints are selected as $\hat{x}_{i,1}^2 + \frac{\hat{x}_{i,2}^2}{(\hat{x}_{i,2} + 1)^2} \leq 0.01$. From equation (30), we find that the maximum step size for the case with set constraints is $T_{\max 1} = 0.038$. From equation (42), the maximum step size for the case with general constraints is $T_{\max 2} = 0.015$. However, we find that the step size can be chosen as 0.1 in our experiment. The DPS algorithm (6) serves as a comparative algorithm as described in (Mai & Abed, 2019). For this algorithm, the step-size is set to $\frac{1}{50(t+1)}$ under circle graph case, and to $\frac{1}{500(t+1)}$ under random graph and complete graph cases.

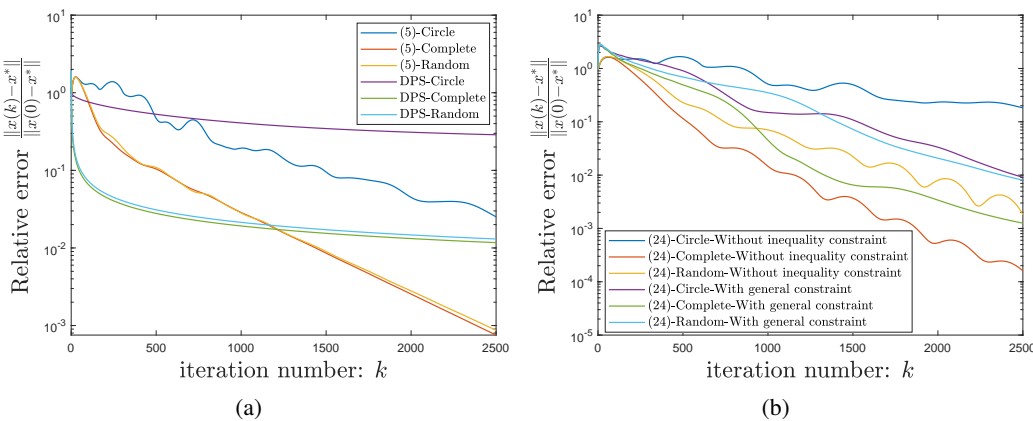

Figure 1: The trajectory of residual. (a) case with set constraints. (b) case with general constraints.

As shown in (a) of Figure 1, compared to the DSP algorithm, our method with constant step sizes may converge more slowly in the initial iterations. However, as the iterations proceed, it can maintain a constant convergence rate. Under the same step size conditions, increasing network connectivity is beneficial for improving the algorithm's convergence speed. For a graph with low network connectivity, such as a circle graph, the DSP algorithm may converge extremely slowly compared to our algorithm. In (b) of Figure 1, we conduct two cases: one with inequality constraints and another without. In the case without inequality constraints, where only equality constraints are introduced, (12) indicates that greater network connectivity benefits the convergence speed. Conversely, in the case with inequality constraints, lower network connectivity is advantageous for convergence speed because the optimal solution lies on the boundary of the feasible region and is dominated by the constraints. The introduction of inequality constraints slows down the convergence speed, as shown in (b) of Figure 1.

## 6 CONCLUSION

In this paper, we have investigated the distributed constrained optimal consensus problem for multi-agent systems under a directed graph. By employing the sliding mode method, we propose a projection-based distributed constrained optimal consensus algorithm that utilizes only relative state information for the optimization problem with set constraints. Furthermore, based on this framework, we have explored the distributed optimization problem with general constraints. The simulation results have illustrated that all agents can converge to the optimal solution with the proposed algorithm.

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

# A APPENDIX

## A.1 PROOF OF THEOREM 3.6

For equation (10), consider the following Lyapunov function candidate $V_1 = V_{1,1} + V_{1,2}$, where

$$V_{1,1}(k) = -\sum_{i=1}^{n}\left[f_i(x_i(k)) - f_i(x^*) - (x_i(k) - x^*)^T\nabla f_i(x_i(k))\right] - \frac{1}{2}\sum_{i=1}^{n}\alpha_i^{-1}\tilde{x}_i^T(k)\tilde{x}_i(k),$$

$$V_{1,2}(k) = \frac{1}{2}\sum_{i=1}^{n}\alpha_i^{-1}\left[(s_i(k) - x^*)^T(s_i(k) - x^*) - (s_i(k) - \mu_i(k))^T(s_i(k) - \mu_i(k))\right]. \quad (20)$$

Let $\tilde{V}_{1,1}(k) = V_{1,1}(k+1) - V_{1,1}(k)$. Taking the difference of the Lyapunov function $V_{1,1}(k)$ yields

$$\tilde{V}_{1,1}(k) = \sum_{i=1}^{n}[-f_i(x_i(k+1)) + f_i(x^*) + \tilde{x}_i^T(k+1)\nabla f_i(x_i(k+1)) - \frac{1}{2\alpha_i}\tilde{x}_i^T(k+1)\tilde{x}_i(k+1)$$

$$+ f_i(x_i(k)) - f_i(x^*) - \tilde{x}_i^T(k)\nabla f_i(x_i(k)) + \frac{1}{2\alpha_i}\tilde{x}_i^T(k)\tilde{x}_i(k)]$$

$$= \sum_{i=1}^{n}[-f_i(x_i(k+1)) + f_i(x_i(k)) + \tilde{x}_i^T(k+1)\nabla f_i(x_i(k+1)) - \tilde{x}_i^T(k)\nabla f_i(x_i(k))$$

$$+ \frac{1}{2\alpha_i}(x_i(k+1) - x_i(k))^T(x_i(k+1) - x_i(k)) - \frac{1}{\alpha_i}\tilde{x}_i(k+1)^T(x_i(k+1) - x_i(k))]$$

$$= \sum_{i=1}^{n}[f_i(x_i(k)) - f_i(x_i(k+1)) - \nabla f_i(x_i(k))(x_i(k) - x_i(k+1))$$

$$+ \tilde{x}_i^T(k+1)(\nabla f_i(x_i(k+1)) - \nabla f_i(x_i(k)))$$

$$+ \frac{1}{2\alpha_i}(\tilde{x}_i(k+1) - \tilde{x}_i(k))^T(\tilde{x}_i(k+1) - \tilde{x}_i(k)) - \frac{1}{\alpha_i}\tilde{x}_i(k+1)^T(\tilde{x}_i(k+1) - \tilde{x}_i(k))]. \quad (21)$$

Note that for any $a \in \mathbb{R}^p$, $\exists \bar{x}_i(k) \in (x_i(k), x_i(k+1))$, and $\exists H_i(k) \in \partial^2 f_i(\bar{x}_i(k))$ such that

$$a^T[h_i(k+1) - h_i(k)] = a^T H_i(k)(\tilde{x}_i(k+1) - \tilde{x}_i(k))$$

$$= Ta^T H_i(k)(\tilde{y}_i(k) - \tilde{x}_i(k)), \quad (22)$$

where $\bar{x}_i(k)$ is weight average of $x_i(k)$ and $x_i(k-1)$, $H_i(k) \in f_i(\bar{x}_i(k))$. By using equation (22), we have

$$\tilde{V}_{1,1}(k) \leq \sum_{i=1}^{n}[-\frac{1}{2}m_i\|x_i(k+1) - x_i(k)\|^2 + \tilde{x}_i^T(k+1)H_i(k)(x_i(k+1) - x_i(k))$$

$$+ \frac{1}{2\alpha_i}\|x_i(k+1) - x_i(k)\|^2 - \frac{1}{\alpha_i}\tilde{x}_i(k+1)^T(x_i(k+1) - x_i(k))]. \quad (23)$$

Substituting equation (15) into equation (23) yields

$$
\begin{aligned}
\tilde{V}_{1,1}(k) \leq & \sum_{i=1}^{n} [-\frac{1}{2}(m_i - \frac{1}{\alpha_i})\|\tilde{x}_i(k+1) - \tilde{x}_i(k)\|^2 - T(\tilde{x}_i(k) - T(\tilde{x}_i(k) - \tilde{\mu}_i(k)))(H_i(k) \\
& - \frac{1}{\alpha_i})(\tilde{x}_i(k) - \tilde{\mu}_i(k))] \\
\leq & \sum_{i=1}^{n} [-\frac{1}{2}(m_i - \frac{1}{\alpha_i})\|\tilde{x}_i(k+1) - \tilde{x}_i(k)\|^2 - T\tilde{x}_i(k)^T(H_i(k) - \frac{1}{\alpha_i})(\tilde{x}_i(k) - \tilde{\mu}_i(k)) \\
& + T^2(\tilde{x}_i(k) - \tilde{\mu}_i(k))(H_i(k) - \frac{1}{\alpha_i})(\tilde{x}_i(k) - \tilde{\mu}_i(k))] \\
\leq & T^2(M_i - \frac{1}{2}m_i - \frac{1}{2\alpha_i})\|\tilde{x}_i(k) - \tilde{\mu}_i(k))\|^2 - T\tilde{x}_i(k)^T(H_i(k) - \frac{1}{\alpha_i})(\tilde{x}_i(k) - \tilde{\mu}_i(k)).
\end{aligned}
\tag{24}
$$

The difference of $V_{1,1}$ satisfies

$$
\begin{aligned}
\tilde{V}_{1,1}(k) \leq & \sum_{i=1}^{n} T^2(M_i - \frac{1}{2}m_i - \frac{1}{2\alpha_i})\|x_i(k) - \tilde{\mu}_i(k))\|^2 \\
& - \sum_{i=1}^{n} T\tilde{x}_i(k)^T(H_i(\bar{x}_i) - \frac{1}{\alpha_i})(\tilde{x}_i(k) - \tilde{\mu}_i(k)).
\end{aligned}
\tag{25}
$$

Let $\tilde{V}_{1,2}(k) = V_{1,2}(k+1) - V_{1,2}(k)$. The difference of $V_{1,2}$ satisfies

$$
\begin{aligned}
\tilde{V}_{1,2}(k) = & \frac{1}{2} \sum_{i=1}^{n} \alpha_i^{-1} [(s_i(k+1) - x^*)^T(s_i(k+1) - x^*) - (s_i(k+1) - \mu_i(k+1))^T(s_i(k+1) \\
& - \mu_i(k+1))] - \frac{1}{2} \sum_{i=1}^{n} \alpha_i^{-1} [(s_i(k) - x^*)^T(s_i(k) - x^*) - (s_i(k) - \mu_i(k))^T(s_i(k) \\
& - \mu_i(k))] \\
= & \frac{1}{2} \sum_{i=1}^{n} \alpha_i^{-1} [(s_i(k+1) - x^*)^T(s_i(k+1) - x^*) - (s_i(k+1) - \mu_i(k+1))^T(s_i(k+1) \\
& - \mu_i(k+1)) + (s_i(k) - \mu_i(k+1))^T(s_i(k) - \mu_i(k+1))] - \frac{1}{2} \sum_{i=1}^{n} \alpha_i^{-1} [(s_i(k) \\
& - x^*)^T(s_i(k) - x^*) - (s_i(k) - \mu_i(k))^T(s_i(k) - \mu_i(k)) - (s_i(k) - \mu_i(k+1))^T(s_i(k) \\
& - \mu_i(k+1))] \\
= & \frac{1}{2} \sum_{i=1}^{n} \alpha_i^{-1} [-(s_i(k+1) - s_i(k))^T(s_i(k+1) - s_i(k)) + 2(s_i(k+1) \\
& - s_i(k))^T(s_i(k+1) - x^*) + (s_i(k+1) - s_i(k))^T(s_i(k+1) - s_i(k)) - 2(s_i(k+1) \\
& - s_i(k))^T(s_i(k+1) - \mu_i(k+1)) + (s_i(k) - \mu_i(k))^T(s_i(k) - \mu_i(k)) - (s_i(k) \\
& - \mu_i(k+1))^T(s_i(k) - \mu_i(k+1))] \\
\leq & \frac{1}{2} \sum_{i=1}^{n} \alpha_i^{-1} [2(s_i(k+1) - s_i(k))^T(\mu_i(k+1) - x^*) - (\mu_i(k+1) - \mu_i(k))^T(\mu_i(k+1) \\
& - \mu_i(k))]
\end{aligned}
\tag{26}
$$

Substituting equation (9) into equation (26), we have

$$
\begin{aligned}
\tilde{V}_{1,2}(k) \leq & [T(\tilde{\mu}(k) - \tilde{x}(k)) - T\alpha\mathcal{L}\tilde{\mu}(k) - \alpha(h(k+1) - h(k))]^T \alpha^{-1}(\tilde{\mu}(k) - \tilde{\mu}(k) \\
& + \tilde{\mu}(k+1)) - \frac{1}{2}(\tilde{\mu}(k+1) - \tilde{\mu}(k))^T \alpha^{-1}(\tilde{\mu}(k+1) - \tilde{\mu}(k))
\end{aligned}
\tag{27}
$$

By using young's inequality,

$$
\begin{aligned}
\tilde{V}_{1,2}(k) \leq &\left[T(\tilde{\mu}(k) - x(k)) - T\alpha\mathcal{L}\tilde{\mu}(k) - \alpha(h(k+1) - h(k))\right]^T \alpha^{-1}(\tilde{\mu}(k) - \tilde{\mu}(k) \\
&+ \tilde{\mu}(k+1)) - \frac{1}{2}(\tilde{\mu}_i(k+1) - \tilde{\mu}_i(k))^T(\tilde{\mu}(k+1) - \tilde{\mu}(k)) \\
\leq &- T\tilde{\mu}^T(k)\mathcal{L}\tilde{\mu}(k) - T\tilde{\mu}^T(k)(H(k) - \alpha^{-1})(\tilde{\mu}(k) - \tilde{x}(k)) + \frac{1}{2}\big[T(\tilde{\mu}(k) - \tilde{x}(k)) \\
&- T\alpha\mathcal{L}\tilde{\mu}(k) - T\alpha H(k)(\tilde{\mu}(k) - \tilde{x}(k))\big]^T \alpha^{-1}\big[T(\tilde{\mu}(k) - \tilde{x}(k)) - T\alpha\mathcal{L}\tilde{\mu}(k) \\
&- T\alpha H(k)(\tilde{\mu}(k) - \tilde{x}(k))\big] - \frac{1}{2}(\tilde{\mu}(k+1) - \tilde{\mu}(k))^T(\tilde{\mu}(k+1) - \tilde{\mu}(k)) \\
\leq &- T\tilde{\mu}^T(k)(H(k) - \alpha^{-1})(\tilde{\mu}(k) - \tilde{x}(k)) - (T\lambda_{\min}(\tilde{\mathcal{L}}) - \bar{\alpha}T^2\|\mathcal{R}\mathcal{L}\mathcal{R}^T\|^2)\|\mathcal{R}_{\otimes}\tilde{\mu}\| \\
&+ \sum_{i=1}^n \frac{T^2}{\alpha_i}(1 - M_i\alpha_i)^2\|\tilde{\mu}_i(k) - \tilde{x}_i(k)\|^2.
\end{aligned}
\tag{28}
$$

Let $\tilde{V}_1(k) = V_1(k+1) - V_1(k)$. The difference of the Lyapunov function $V_1$ satisfies

$$
\begin{aligned}
\tilde{V}_1(k) \leq &- T(\tilde{\mu}(k) - \tilde{x}(k))^T(H(k) - \alpha^{-1})(\tilde{\mu}(k) - \tilde{x}(k)) - (T\lambda_{\min}(\tilde{\mathcal{L}}) \\
&- \bar{\alpha}T^2\|\mathcal{R}\mathcal{L}\mathcal{R}^T\|^2)\|\mathcal{R}_{\otimes}\tilde{\mu}\|^2 + \sum_{i=1}^n \frac{T^2}{\alpha_i}(1 - M_i\alpha_i)^2\|\tilde{\mu}_i(k) - \tilde{x}_i(k)\|^2 \\
&+ \sum_{i=1}^n T^2(M_i - \frac{1}{2}m_i - \frac{1}{2\alpha_i})\|\tilde{x}_i(k) - \tilde{\mu}_i(k))\|^2 \\
\leq &- T\sum_{i=1}^n [m_i - \frac{1}{\alpha_i} - T(M_i - \frac{1}{2}m_i - \frac{1}{2\alpha_i} + \frac{1}{\alpha_i}(1 - M_i\alpha_i)^2)]\|\tilde{x}_i(k) - \tilde{\mu}_i(k))\|^2 \\
&- T(\lambda_{\min}(\mathcal{R}\tilde{\mathcal{L}}\mathcal{R}^T) - \bar{\alpha}T\|\mathcal{R}\mathcal{L}\mathcal{R}^T\|^2)\|\mathcal{R}_{\otimes}\tilde{\mu}\|^2.
\end{aligned}
\tag{29}
$$

Let $k_{Ti} = \frac{m_i - \frac{1}{\alpha_i}}{M_i - \frac{1}{2}m_i - \frac{1}{2\alpha_i} + \frac{1}{\alpha_i}(1 - M_i\alpha_i)^2}$, Choosing step size $T$ such that

$$
T \leq \min_{i=1,\ldots,n}(k_{Ti}, \frac{\lambda_{\min}(\mathcal{R}\tilde{\mathcal{L}}\mathcal{R}^T)}{\bar{\alpha}\|\mathcal{R}\mathcal{L}\mathcal{R}^T\|^2}).
\tag{30}
$$

Choosing $\beta = \min_{i=1,\ldots,n}(T[m_i - \frac{1}{\alpha_i} - T(M_i - \frac{1}{2}m_i - \frac{1}{2\alpha_i} + \frac{1}{\alpha_i}(1 - M_i\alpha_i)^2)], T(\lambda_{\min}(\mathcal{R}\tilde{\mathcal{L}}\mathcal{R}^T) - \bar{\alpha}T\|\mathcal{R}\mathcal{L}\mathcal{R}^T\|^2))$. From the definition of step size, we have $\beta > 0$. Therefore, Then, equation (29) can be written as

$$
\tilde{V}_1(k) \leq -\beta\|\tilde{x}(k) - \tilde{\mu}(k))\|^2 - \beta\|\mathcal{R}_{\otimes}\tilde{\mu}\|^2.
\tag{31}
$$

According to Lemma 2.2, we have $\{V_1(k)\}$ is monotonously decreasing and $\lim_{k\to\infty} V_1(k) \in \mathbb{L}_\infty$. According to Cauchy criterion, $\lim_{k\to\infty}(V_1(k+1) - V_1(k)) = \mathbf{0}_{np}$, $\lim_{k\to\infty}(\tilde{x}(k) - \tilde{\mu}(k)) = \mathbf{0}_{np}$, $\lim_{k\to\infty}\mathcal{R}_{\otimes}\tilde{\mu}(k) = \mathbf{0}_{(n-1)p}$. Following the same procedure in *Equilibrium analysis*, we have $\lim_{k\to\infty} x_i(k) = x^*$. From equation (31), we have $\lim_{k\to\infty} \frac{1}{k}\sum_{t=1}^k \|\tilde{x}(t) - \tilde{\mu}(t))\|^2$, $\frac{1}{k}\sum_{t=1}^k \|\mathcal{R}_{\otimes}\tilde{x}(t)\|^2$, $\frac{1}{k}\sum_{t=1}^k \|A_i\tilde{\mu}_i(t)\|^2 < \infty$. From equation (7), $\lim_{k\to\infty} \frac{1}{k}\sum_{t=1}^k \|\tilde{x}_i(t+1) - \tilde{x}_i(t))\|^2 < \infty$. Finally, the first-order optimality residuals converge with a rate of $\mathcal{O}(\frac{1}{k})$, which completes the proof.

## A.2 PROOF OF THEOREM 4.4

For equation (19), consider the Lyapunov function candidate $V_2 = V_{2,1} + V_{2,2}$ with

$$
V_{2,1} = \frac{1}{2} \sum_{i=1}^{n} \alpha_i^{-1} \big[ (y_{\mu i}(k) - \bar{\mu})^T (y_{\mu i}(k) - \bar{\mu}) - (y_{\mu i}(k) - \mu_i(k))^T (y_{\mu i}(k) - \mu_i(k)) \big].
$$

$$
V_{2,2} = - \sum_{i=1}^{n} \big[ f_i(x_i(k)) - f_i(\bar{\mu}) - (x_i(k) - \bar{\mu})^T \nabla f_i(x_i(k)) \big] - \frac{1}{2} \sum_{i=1}^{n} \alpha_i^{-1} \tilde{x}_i^T(k) \tilde{x}_i(k)
$$

$$
+ \sum_{i=1}^{n} \sum_{k=1}^{p_{g_i}} \gamma_{ik}(k)(-g_{ik}(x_i(k)) + \nabla g_{ik}^T(x_i(k)) \tilde{x}_i(k)). \tag{32}
$$

Building on Lemma 2.3, we can establish the positive definiteness of $V_{2,1}$ through the inequality $V_{2,1} \geq \frac{1}{2} \sum_{i=1}^{n} \alpha_i^{-1} \|\tilde{\mu}_i(k)\|^2$. From the strong convexity of $f_i$, we have $-[f_i(x_i(k)) - f_i(\bar{\mu}) - \nabla f_i^T(x_i(k))(x_i(k) - \bar{\mu})] \geq \frac{m_i}{2} \|x_i(k) - \bar{\mu}\|^2$. Therefore, we conclude that $V_{2,2} \geq \left( \frac{m_i}{2} - \frac{1}{2\alpha_i} \right) \tilde{x}^T \tilde{x}$. By choosing $\alpha_i > \frac{1}{m_i}$, we can derive the positive definiteness of $V_{2,2}$ with respect to $\tilde{x}^T \tilde{x}$. From the convexity of $g_{ik}(x_i(k))$, we have $-g_{ik}(x_i(k)) + \nabla g_{ik}^T(x_i(k)) \tilde{x}_i(k) \geq -g_{ik}(\bar{\mu}) > 0$. From the above inequalities, we derive that $\sum_{i=1}^{n} \sum_{k=1}^{p_{g_i}} \gamma_{ik}(k)(-g_{ik}(x_i(k)) + \nabla g_{ik}^T \tilde{x}_i(k)) \geq -\sum_{i=1}^{n} \sum_{k=1}^{p_{g_i}} \gamma_{ik}(k) g_{ik}(\bar{\mu})$ and $V_{2,2}$ is positive definite with respect to $\gamma_{ik}$. In summary, $V_{2,2}$ is positive definite with respect to $\tilde{x}_i^T(k) \tilde{x}_i(k)$ and $\gamma_{ik}(k)$ within our error definition.

Let $\tilde{V}_{2,1}(k) = V_{2,1}(k+1) - V_{2,1}(k)$. Based on Lemma 2.3, the difference of $V_{2,1}$ satisfies

$$
\begin{aligned}
\tilde{V}_{2,1}(k) \leq & \frac{1}{2} \sum_{i=1}^{n} \alpha_i^{-1} \big[ 2(y_{\mu i}(k+1) - y_{\mu i}(k))^T (\mu_i(k+1) - x^*) - (\mu_i(k+1) \\
& - \mu_i(k))^T (\mu_i(k+1) - \mu_i(k)) \big] \\
\leq & \frac{1}{2} \sum_{i=1}^{n} \alpha_i^{-1} \big[ 2(y_{\mu i}(k+1) - y_{\mu i}(k))^T (\mu_i(k) - x^*) + (y_{\mu i}(k+1) \\
& - y_{\mu i}(k))^T (y_{\mu i}(k+1) - y_{\mu i}(k)) \big].
\end{aligned} \tag{33}
$$

Substituting equation (17) into equation (33) yields

$$
\begin{aligned}
\tilde{V}_{2,1}(k) \leq & \sum_{i=1}^{n} \alpha_i^{-1} \Big\{ T(\tilde{\mu}_i(k) - \tilde{x}_i(k)) - \alpha_i \Big[ T \sum_{j=1}^{n} a_{ij}(\tilde{\mu}_i(k) - \tilde{\mu}_j(k)) + \nabla f_i(x_i(k+1)) \\
& - \nabla f_i(x_i(k)) + \sum_{k=1}^{p_{g_i}} \alpha_{\gamma i} \gamma_{ik}(k+1) \nabla g_{ik}(x_i(k+1)) - \sum_{k=1}^{p_{g_i}} \alpha_{\gamma i} \gamma_{ik}(k) \nabla g_{ik}(x_i(k)) \\
& + T \alpha_{\nu i} A_i^T A_i \tilde{\mu}_i(k) \Big] \Big\}^T \tilde{\mu}_i(k) + \frac{1}{2} \sum_{i=1}^{n} \alpha_i^{-1} \| T(\mu_i(k) - x_i(k)) - \alpha_i \Big[ T \sum_{j=1}^{n} a_{ij}(\mu_i(k) \\
& - \mu_j(k)) + \nabla f_i(x_i(k+1)) - \nabla f_i(x_i(k)) + \sum_{k=1}^{p_{g_i}} \alpha_{\gamma i} \gamma_{ik}(k+1) \nabla g_{ik}(x_i(k+1)) \\
& - \sum_{k=1}^{p_{g_i}} \alpha_{\gamma i} \gamma_{ik}(k) \nabla g_{ik}(x_i(k)) + T \alpha_{\nu i} A_i^T A_i \tilde{\mu}_i(k) \Big] \|^2.
\end{aligned} \tag{34}
$$

By using equation (18) and Young's inequality, we have

$$
\begin{aligned}
\tilde{V}_{2,1}(k) \leq & -T \sum_{i=1}^{n} \tilde{\mu}_i^T(k)(m_i - \alpha_i^{-1})(\tilde{\mu}_i(k) - \tilde{x}_i(k)) - T \sum_{i=1}^{n} \tilde{\mu}_i^T \bar{\gamma}_{ik} H_{gi}(k)(\tilde{\mu}_i(k) - \tilde{x}_i(k)) \\
& - T \sum_{i=1}^{n} \tilde{\mu}_i^T \nabla g_{ik}(\bar{x}_i(k)) \Big\{ -\alpha_{\gamma i} \gamma_{ik}(k) + \alpha_{\gamma i} \big[ \gamma_{ik}(k) + g_{ik}(x_i(k)) \\
& - \nabla g_{ik}^T(x_i(k))(\tilde{x}_i(k) - \tilde{\mu}_i(k)) \big]^+ \Big\} - T \sum_{i=1}^{n} \alpha_{\nu i} \tilde{\mu}_i^T(k) A_i^T A_i \tilde{\mu}_i(k) \\
& + 2T \sum_{i=1}^{n} \alpha_i^{-1}(\alpha_i M_i - 1)^2 \|\tilde{\mu}_i(k) - \tilde{x}_i(k)\|^2 - T(\lambda_{\min}(\mathcal{R}\tilde{\mathcal{L}}\mathcal{R}^T) \\
& - 2\bar{\alpha} T \|\mathcal{R}\mathcal{L}\mathcal{R}^T\|^2) \|\mathcal{R}_{\otimes} \tilde{\mu}\|^2 + 2T^2 \sum_{i=1}^{n} \lambda_{\max}(A_i^T A_i) \alpha_{\nu i}^2 \tilde{\mu}_i^T(k) A_i^T A_i \tilde{\mu}_i(k) \\
& + 4 \sum_{i=1}^{n} \bar{g}_{ik}^2 \| \sum_{k=1}^{p_{g_i}} \alpha_{\gamma i}(\gamma_{ik}(k+1) - \gamma_{ik}(k))\|^2 + 4 \sum_{i=1}^{n} \bar{\gamma}_{ik}^2 \| \sum_{k=1}^{p_{g_i}} \alpha_{\gamma i} \nabla g_{ik}(x_i(k+1)) \\
& - \nabla g_{ik}(x_i(k))\|^2.
\end{aligned}
\tag{35}
$$

Let $\tilde{V}_{2,2}(k) = V_{2,2}(k+1) - V_{2,2}(k)$. Taking the difference of $V_{2,2}$ yields

$$
\begin{aligned}
\tilde{V}_{2,2}(k) \leq & \sum_{i=1}^{n} T^2(M_i - \frac{1}{2}m_i - \frac{1}{2\alpha_i})\|\tilde{x}_i(k) - \tilde{\mu}_i(k)\|^2 - \sum_{i=1}^{n} T \tilde{x}_i(k)^T (H_i(k) - \frac{1}{\alpha_i})(\tilde{x}_i(k) \\
& - \tilde{\mu}_i(k)) + \sum_{i=1}^{n} \sum_{k=1}^{p_{g_i}} \gamma_{ik}(k+1)(-g_{ik}(x_i(k+1)) + \nabla g_{ik}^T(x_i(k+1))\tilde{x}_i(k+1)) \\
& - \sum_{i=1}^{n} \sum_{k=1}^{p_{g_i}} \gamma_{ik}(k)(-g_{ik}(x_i(k)) + \nabla g_{ik}^T(x_i(k))\tilde{x}_i(k)).
\end{aligned}
$$

Note that

$$
\begin{aligned}
& \sum_{i=1}^{n} \sum_{k=1}^{p_{g_i}} \Big[ \gamma_{ik}(k+1)(-g_{ik}(x_i(k+1)) + \nabla g_{ik}^T(x_i(k+1))\tilde{x}_i(k+1)) - \gamma_{ik}(k)(-g_{ik}(x_i(k)) \\
& \quad + \nabla g_{ik}^T(x_i(k))\tilde{x}_i(k)) \Big] \\
\leq & \sum_{i=1}^{n} \sum_{k=1}^{p_{g_i}} \Big[ \bar{\gamma}_{ik} H_{gik}(k)(\tilde{x}_i(k+1) - \tilde{x}_i(k)) - (\gamma_{ik}(k+1) - \gamma_{ik}(k))(-g_{ik}(\bar{x}_i(k)) \\
& \quad + \nabla g_{ik}^T(\bar{x}_i(k))\tilde{\tilde{x}}_i(k)) \Big] \\
= & \sum_{i=1}^{n} \sum_{k=1}^{p_{g_i}} \Big[ T H_{gik}(k)(-\tilde{x}_i(k) + \tilde{\mu}_i(k)) - (\gamma_{ik}(k+1) - \gamma_{ik}(k))(-g_{ik}(\bar{x}_i(k)) \\
& \quad + \nabla g_{ik}^T(\bar{x}_i(k))\tilde{\tilde{x}}_i(k)) \Big].
\end{aligned}
\tag{36}
$$

Let $\tilde{V}_2(k) = V_2(k+1) - V_2(k)$. Then, we have

$$
\begin{aligned}
\tilde{V}_2(k) \leq &-T \sum_{i=1}^{n}[m_i - \frac{1}{\alpha_i} + \bar{\gamma}_{ik}\lambda_{\min}(H_{gi}(k)) - T(M_i - \frac{1}{2}m_i - \frac{1}{2\alpha_i} + \frac{2}{\alpha_i}(1 - M_i\alpha_i)^2 \\
&+ 4\alpha_{\gamma i}^2 p_{g_i} \sum_{k=1}^{p_{g_i}} \bar{\gamma}_{ik}^2 M_{gik}^2)]\|\tilde{x}_i(k) - \tilde{\mu}_i(k))\|^2 - T(\lambda_{\min}(\mathcal{R}\tilde{\mathcal{L}}\mathcal{R}^T) \\
&- 2\bar{\alpha}T\|\mathcal{R}\mathcal{L}\mathcal{R}^T\|^2)\|\mathcal{R}_{\otimes}\tilde{\mu}\|^2 - T\sum_{i=1}^{n}(\alpha_{\nu i} - 2T\lambda_{\max}(A_i^T A_i)\alpha_{\nu i}^2)\tilde{\mu}_i^T(k)A_i^T A_i\tilde{\mu}_i(k) \\
&- \sum_{i=1}^{n}\sum_{k=1}^{p_{g_i}}(\gamma_{ik}(k+1) - \gamma_{ik}(k))(-g_{ik}(\bar{x}_i(k)) + \nabla g_{ik}^T(\bar{x}_i(k))(\tilde{\bar{x}}_i(k) - \tilde{\mu}_i(k))) \\
&+ 4\sum_{i=1}^{n}\alpha_{\gamma i}^2 p_{g_i} \sum_{k=1}^{p_{g_i}} \bar{g}_{ik}^2\|(\gamma_{ik}(k+1) - \gamma_{ik}(k))\|^2.
\end{aligned}
\tag{37}
$$

Note that

$$
\begin{aligned}
&\sum_{i=1}^{n}\sum_{k=1}^{p_{g_i}}(\gamma_{ik}(k+1) - \gamma_{ik}(k))(-g_{ik}(\bar{x}_i(k)) + \nabla g_{ik}^T(\bar{x}_i(k))(\tilde{\bar{x}}_i(k) - \tilde{\mu}_i(k))) \\
&= \sum_{i=1}^{n}\sum_{k=1}^{p_{g_i}}(\gamma_{ik}(k+1) - \gamma_{ik}(k))\big[ - g_{ik}(x_i(k)) + \nabla g_{ik}^T(x_i(k))(\tilde{x}_i(k) - \tilde{\mu}_i(k)) \\
&+ g_{ik}(x_i(k)) - g_{ik}(\bar{x}_i(k)) - \nabla g_{ik}^T(\bar{x}_i(k))(x_i(k) - \bar{x}_i(k)) - (\nabla g_{ik}(x_i(k)) \\
&- \nabla g_{ik}(\bar{x}_i(k)))^T(\tilde{x}_i(k) - \tilde{\mu}_i(k))\big].
\end{aligned}
\tag{38}
$$

We next prove the negative definiteness of term $(\gamma_{ik}(k+1) - \gamma_{ik}(k))(-g_{ik}(x_i(k)) + \nabla g_{ik}^T(x_i(k))(\tilde{x}_i(k) - \tilde{\mu}_i(k)))$.

$$
\begin{aligned}
&(\gamma_{ik}(k+1) - \gamma_{ik}(k))(-g_{ik}(x_i(k)) + \nabla g_{ik}^T(x_i(k))(\tilde{x}_i(k) - \tilde{\mu}_i(k))) \\
&= \begin{cases} -T^{-1}\alpha_{\gamma i}^{-1}(\gamma_{ik}(k+1) - \gamma_{ik}(k))^2, & \text{if } (\gamma_{ik} + g_{ik}(x_i) - \nabla g_{ik}^T(x_i)(x_i - \mu_i)) \geq 0. \\ (\gamma_{ik}(k+1) - \gamma_{ik}(k))(-g_{ik}(x_i(k)) + \nabla g_{ik}^T(x_i(k))(\tilde{x}_i(k) - \tilde{\mu}_i(k))), & \text{else.} \end{cases}
\end{aligned}
$$

For the case $(\gamma_{ik}(k+1) - \gamma_{ik}(k))(-g_{ik}(x_i(k)) + \nabla g_{ik}^T(x_i(k))(\tilde{x}_i(k) - \tilde{\mu}_i(k))) < 0$, we have $\gamma_{ik}(k+1) - \gamma_{ik}(k) = -\alpha_{\gamma i}\gamma_{ik}(k) \leq 0$ and $-g_{ik}(x_i(k)) + \nabla g_{ik}^T(x_i(k))(\tilde{x}_i(k) - \tilde{\mu}_i(k)) > \gamma_{ik}(k) = -\alpha_{\gamma i}^{-1}(\gamma_{ik}(k+1) - \gamma_{ik}(k))$. Therefore, $(\gamma_{ik}(k+1) - \gamma_{ik}(k))(-g_{ik}(x_i(k)) + \nabla g_{ik}^T(x_i(k))(\tilde{x}_i(k) - \tilde{\mu}_i(k))) \leq -\alpha_{\gamma i}^{-1}(\gamma_{ik}(k+1) - \gamma_{ik}(k))^2$. Therefore, $(\gamma_{ik}(k+1) - \gamma_{ik}(k))(-g_{ik}(x_i(k)) + \nabla g_{ik}^T(x_i(k))(\tilde{x}_i(k) - \tilde{\mu}_i(k))) \leq -T^{-1}\alpha_{\gamma i}^{-1}(\gamma_{ik}(k+1) - \gamma_{ik}(k))^2$ under any case. Therefore, equation (38) can be written as

$$
\begin{aligned}
&\sum_{i=1}^{n}\sum_{k=1}^{p_{g_i}}(\gamma_{ik}(k+1) - \gamma_{ik}(k))(-g_{ik}(\bar{x}_i(k)) + \nabla g_{ik}^T(\bar{x}_i(k))(\tilde{\bar{x}}_i(k) - \tilde{\mu}_i(k))) \\
&\leq -\sum_{i=1}^{n}\sum_{k=1}^{p_{g_i}} T^{-1}\alpha_{\gamma i}^{-1}(\gamma_{ik}(k+1) - \gamma_{ik}(k))^2 + \sum_{i=1}^{n}\sum_{k=1}^{p_{g_i}}(\gamma_{ik}(k+1) - \gamma_{ik}(k))(g_{ik}(x_i(k)) \\
&- g_{ik}(\bar{x}_i(k)) - \nabla g_{ik}^T(x_i(k))(x_i(k) - \bar{x}_i(k))) \\
&\leq -\frac{1}{2}\sum_{i=1}^{n}\sum_{k=1}^{p_{g_i}} T^{-1}\alpha_{\gamma i}^{-1}(\gamma_{ik}(k+1) - \gamma_{ik}(k))^2 + \frac{1}{2}\sum_{i=1}^{n}\sum_{k=1}^{p_{g_i}} T\alpha_{\gamma i}\|g_{ik}(x_i(k)) - g_{ik}(\bar{x}_i(k)) \\
&- \nabla g_{ik}^T(x_i(k))(x_i(k) - \bar{x}_i(k))\|^2.
\end{aligned}
\tag{39}
$$

From Assumption 3.2, $g_{ik}(x_i(k)) \leq g_{ik}(\bar{x}_i(k)) + \nabla g_{ik}^T(\bar{x}_i(k))(x_i(k) - \bar{x}_i(k)) + \frac{M_{gi}}{2}\|x_i(k) - \bar{x}_i(k)\|^2$. Therefore, we have

$$\sum_{i=1}^{n}\sum_{k=1}^{p_{g_i}}(\gamma_{ik}(k+1) - \gamma_{ik}(k))(-g_{ik}(\bar{x}_i(k)) + \nabla g_{ik}^T(\bar{x}_i(k))(\tilde{\bar{x}}_i(k) - \tilde{\mu}_i(k)))$$

$$\leq -\frac{1}{2}\sum_{i=1}^{n}\sum_{k=1}^{p_{g_i}}T^{-1}\alpha_{\gamma i}^{-1}(\gamma_{ik}(k+1) - \gamma_{ik}(k))^2 + \frac{1}{4}\sum_{i=1}^{n}\sum_{k=1}^{p_{g_i}}T\alpha_{\gamma i}M_{gi}\|x_i(k) - \bar{x}_i(k)\|^2$$

$$\leq -\frac{1}{2}\sum_{i=1}^{n}\sum_{k=1}^{p_{g_i}}T^{-1}\alpha_{\gamma i}^{-1}(\gamma_{ik}(k+1) - \gamma_{ik}(k))^2 + \frac{1}{4}\sum_{i=1}^{n}\sum_{k=1}^{p_{g_i}}T\alpha_{\gamma i}M_{gi}\|x_i(k+1) - x_i(k)\|^2$$

$$= -\frac{1}{2}\sum_{i=1}^{n}\sum_{k=1}^{p_{g_i}}T^{-1}\alpha_{\gamma i}^{-1}(\gamma_{ik}(k+1) - \gamma_{ik}(k))^2 + \frac{1}{4}T^3\sum_{i=1}^{n}\sum_{k=1}^{p_{g_i}}\alpha_{\gamma i}^2 M_{gi}\|\tilde{x}_i(k) - \tilde{\mu}_i(k)\|^2. \quad (40)$$

Substituting equation (40) into equation (37) yields

$$\tilde{V}_2(k) \leq -T\sum_{i=1}^{n}[m_i - \frac{1}{\alpha_i} - T(M_i - \frac{1}{2}m_i - \frac{1}{2\alpha_i} + \frac{2}{\alpha_i}(1 - M_i\alpha_i)^2 + 4\alpha_{\gamma i}^2 p_{g_i}\sum_{k=1}^{p_{g_i}}\bar{\gamma}_{ik}^2 M_{gik}^2$$

$$+ \frac{1}{4}T\sum_{k=1}^{p_{g_i}}\alpha_{\gamma i}^2 M_{gi})]\|\tilde{x}_i(k) - \tilde{\mu}_i(k))\|^2 - T(\lambda_{\min}(\mathcal{R}\tilde{\mathcal{L}}\mathcal{R}^T) - 2\bar{\alpha}T\|\mathcal{R}\mathcal{L}\mathcal{R}^T\|^2)\|\mathcal{R}_{\otimes}\tilde{\mu}\|^2$$

$$- T\sum_{i=1}^{n}\alpha_{\nu i}(1 - 2T\lambda_{\max}(A_i^T A_i)\alpha_{\nu i})\tilde{\mu}_i^T(k)A_i^T A_i\tilde{\mu}_i(k) - T^{-1}\sum_{i=1}^{n}\sum_{k=1}^{p_{g_i}}(\frac{1}{2}\alpha_{\gamma i}^{-1}$$

$$- 4T\alpha_{\gamma i}^2 p_{g_i}\bar{g}_{ik}^2)(\gamma_{ik}(k+1) - \gamma_{ik}(k))^2. \quad (41)$$

Let $k_{Ti} = \frac{m_i - \frac{1}{\alpha_i}}{(M_i - \frac{1}{2}m_i - \frac{1}{2\alpha_i} + \frac{2}{\alpha_i}(1 - M_i\alpha_i)^2 + 4\alpha_{\gamma i}^2 p_{g_i}\sum_{k=1}^{p_{g_i}}\bar{\gamma}_{ik}^2 M_{gik}^2 + \frac{1}{4}\sum_{k=1}^{p_{g_i}}\alpha_{\gamma i}^2 M_{gi})}$, and $k_{\nu} = \min_{i=1,...,n}(\frac{1}{2\lambda_{\max}(A_i^T A_i)\alpha_{\nu i}})$, $k_{\gamma} = \min_{\substack{i=1,...,n \\ k=1,...,p_{g_i}}}(\frac{1}{8\alpha_{\gamma i}^3 p_{g_i}\bar{g}_{ik}^2})$. Choosing step size $T$ such that

$$T \leq \min_{i=1,2,...,n}(k_T, k_{\nu}, k_{\gamma}, \frac{\lambda_{\min}(R\tilde{\mathcal{L}}R^T)}{2\bar{\alpha}\lambda_{\max}^2(R\mathcal{L}R^T)}). \quad (42)$$

Let $\hat{\beta} = \min_{i=1,...,n}(m_i - \frac{1}{\alpha_i} - T(M_i - \frac{1}{2}m_i - \frac{1}{2\alpha_i} + \frac{2}{\alpha_i}(1 - M_i\alpha_i)^2 + 4\alpha_{\gamma i}^2 p_{g_i}\sum_{k=1}^{p_{g_i}}\bar{\gamma}_{ik}^2 M_{gik}^2 + \frac{1}{4}T\sum_{k=1}^{p_{g_i}}\alpha_{\gamma i}^2 M_{gi}), T(\lambda_{\min}(\mathcal{R}\tilde{\mathcal{L}}\mathcal{R}^T) - 2\bar{\alpha}T\|\mathcal{R}\mathcal{L}\mathcal{R}^T\|^2), \frac{1}{2}\alpha_{\gamma i}^{-1} - 4T\alpha_{\gamma i}^2 p_{g_i}\bar{g}_{ik}^2, T\alpha_{\nu i}(1 - 2T\lambda_{\max}(A_i^T A_i)\alpha_{\nu i}))$. According to Assumption 3.3, we can obtain

$$\tilde{V}_2(k) \leq -\hat{\beta}\sum_{i=1}^{n}\|\tilde{x}_i(k) - \tilde{\mu}_i(k))\|^2 - \hat{\beta}\|\mathcal{R}_{\otimes}\tilde{\mu}\|^2 - \hat{\beta}\sum_{i=1}^{n}\tilde{\mu}_i^T(k)A_i^T A_i\tilde{\mu}_i(k)$$

$$- \hat{\beta}\sum_{i=1}^{n}\sum_{k=1}^{p_{g_i}}(\gamma_{ik}(k+1) - \gamma_{ik}(k))^2.$$

From equation (37), we have $\tilde{\mu}_i(k)$, $\tilde{x}_i(k)$, $\gamma_{ik}(k)$ are bounded. According to Lemma 2.2, we have positive sequence $\{V(k)\}$ is monotonously decreasing. By using Cauchy criterion, $\lim_{k\to\infty}(\tilde{x}(k) - \tilde{\mu}(k)) = \mathbf{0}_{np}$, $\lim_{k\to\infty}\mathcal{R}_{\otimes}\tilde{\mu}(k) = \mathbf{0}_{(n-1)p}$, $\lim_{k\to\infty}A_i\tilde{\mu}_i(k) = \mathbf{0}_{p_{bi}}$, $\lim_{k\to\infty}(\gamma_{ik}(k+1) - \gamma_{ik}(k)) = 0$. Following the same procedure in *Equilibrium analysis*, we have $\lim_{k\to\infty}x_i(k) = x^*$. Based on equation (37), we also have $\lim_{k\to\infty}\frac{1}{k}\sum_{t=1}^{k}\|\tilde{x}_i(t) - \tilde{\mu}_i(t))\|^2$, $\frac{1}{k}\sum_{t=1}^{k}\|\mathcal{R}_{\otimes}\tilde{\mu}(t)\|^2$, $\frac{1}{k}\sum_{t=1}^{k}\|\mathcal{R}_{\otimes}\tilde{x}(t)\|^2$, $\frac{1}{k}\sum_{t=1}^{k}\|A_i\tilde{\mu}_i(t)\|^2$, $\frac{1}{k}\sum_{t=1}^{k}(\gamma_{ik}(k+1) - \gamma_{ik}(k))^2 < \infty$. From equation (15), $\lim_{k\to\infty}\frac{1}{k}\sum_{t=1}^{k}\|\tilde{x}_i(t+1) - \tilde{x}_i(t))\|^2 < \infty$. Therefore, we have demonstrated that the first-order optimality residuals converge with a rate of $\mathcal{O}(\frac{1}{k})$, which completes the proof.

