# OpenReview forum: "Distributed Constrained Optimal  Consensus  Under a Directed Graph"
_ICLR.cc/2025/Conference — Submitted to ICLR 2025_

### Official Review · Reviewer_WgMd · 2024-10-18

**Soundness:** 2
**Presentation:** 1
**Contribution:** 2
**Rating:** 3
**Confidence:** 4

**Summary:**

The paper develops algorithms for distributed optimization in directed graphs under agent-available constraints (both set and set + inequalities/equalities).

**Strengths:**

+ Methods with constant step-size for distributed constrained optimization.
+ Connections with the sliding mode technique.

**Weaknesses:**

- Restrictive and inappropriate assumptions: Assumption 2.1 (balanced graph) is restrictive, while Assumption 4.3 is not suitable because $\gamma_{ik}$'s are updated in the algorithm, so their boundedness cannot be assumed for convergence analysis (instead it requires a proof).
- Very poor writing.
- Step-size selection (also in a distributed manner) is not properly discussed.
- The problem chosen for experiments is artificial (as opposed to choosing one with possible applications, e.g., MPC). The step-size parameters were manually tuned for best performance (this is unpractical). There is only one baseline method to compare against.

**Questions:**

Questions:

1. Can you relax the assumption on the graph being balanced? This seems quite restrictive in real applications. Please also compare your contributions with the rich related literature on the subject (e.g., DIAG, push-sum, push-pull, etc.).
2. Can you provide a rigorous proof of boundedness for $\gamma_{ik}(k)$ so as to omit Assumption 4.3? If so, does it apply locally (close to optimality) or globally? Alternatively, can you revise your analysis to avoid this assumption? Please also remove or replace the statement about "continuous experimentation and testing" in lines 407-408 with a more rigorous justification.
3. Can you provide a step-size selection rule that is efficiently computable in a distributed manner? In particular:
a) Can you please explicitly describe how equations (30) and (42) can be computed in a distributed manner, if possible?
b) If not, can you propose an alternative distributed method for step-size selection and analyze its cost?
c) Can you also please provide clear ranges or guidelines for selecting $\alpha_i,\alpha_{v_i},\alpha_{\gamma_{ik}}$ in Theorem 4.4?
4. Can you please
a) provide experiments on a more realistic application problem (such as distributed model predictive control or resource allocation)?
b) compare with more baseline methods (see 1.above for examples)?
c) describe a more systematic and distributedly amenable approach for parameter tuning?

Suggestions:

- Please add the step-size selection rules and convergence rate in the statements of your main theorems.
- I suggest writing Algorithm 1 and Algorithm 2 for the two methods and summarizing the key convergence results in comparison with relevant methods in a table in the main paper. You can compress the background material which is almost 50% of the current paper. There is also a lot of redundancy in the paper, e.g., the equilibrium equations on pages 6 and 8 are obvious.
- The language used is not very standard. The terms distributed optimization and consensus optimization are more widely used (compared to "distributed optimal consensus").
- The early work on the subject can be traced back to the seminal PhD thesis of John Tsitsiklis in the 80s, not just over the last decade or so as you state (line 30).
- Please explain more on the second-order dynamics mentioned in line 46, for completeness.
- Please define what you mention as "relative state" (I believe you mean what is most commonly referred to as a local variable in the literature).
- Theorem 2.2 is a classic result not worth stating.


Editing:

I am very sorry to say that the authors did not even make a minimal effort to proofread their paper before submitting it.
The amount of typos and poor phrasing & writing style are excessive in this paper. Below is a non-exhaustive list:

- line 16: case -> the case.
- line 27: optimization of -> optimization for.
- line 29: construct -> constructing.
- line 45: has linear -> have linear.
- line 54: of incorporating -> to incorporate.
- line 63: primal-dual method -> primal-dual methods.
- line 88: the constant -> constant.
- line 89: By omitting local set constraints -> In the absence of local set constraints.
- line 111: balanced graph -> balanced; additionally, lines 111-112 should be placed after line 119.
- lines 131-133: poor phrasing.
- line 137: spinning -> spanning.
- line 148: delete i.e.,; also mention that $M_l$ depends on $U$.
- line 151: delete superscript ^2 in the definition of Lipschitz continuity.
- line 154: general -> generalized.
- In Definition 2.3, delete "the convex hull of the set".
- line 159: better to use different symbol for matrix, e.g., $H$, since $M$ was used for constants. In general, using capital letters for both scalars and matrices is confusing.
- line 166: projection variable -> variable.
- Using different symbols in Lemma 2.2 and (2) is poor style.
- In line 174, I believe the normal cone is defined for $x\in Omega$ (missing).
- It is customary to write function of two variables as $V(x,y)$ in Lemma 2.3 and thereafter.
- line 192: I do not understand "multiple first-order integrators".
- line 197: by a -> that is.
- line 199: choice of notation $q$ in $f_i(q)$ is awkward.
- line 210 and similar instances (e.g., line 227): equation number suffices, i.e., equation (3) -> (3).
- line 215 and line 323, \subseteq in the normal cones should be \supseteq; also in line 323 $\Omega_i$ is misstyped.
- line 228: . Or -> or.
- line 241: $T$ is not defined (step-size).
- line 258: delete "=" at the end of the line.
- line 271: $y_i$ should be $s_i$.
- In (8), $h_i$ is undefined (gradients).
- line 293: The detail of proof is given -> The details of the proof are given.
- Lemma 4.1: KarushKuhnTucker -> Karush-Kuhn-Tucker.
- line 360: is -> are.
- in (13) and thereafter you use $k$ both for iteration count and inequality counter, i.e., $\gamma_{ik}(k)$; please work to fix the notational issues throughout the paper.
- line 403: add "." at the end.
- I do not understand $\in$ in (17).
- In line 477: this should be the most non-standard way of writing a quadratic I have seen; mention also your choice of A_i in the experiments.
- after Fig. 1: in (a) of Figure 1 -> in Figure 1(a) [same for (b)].
- line 702: young's -> Young's.
- line 743: delete "Then, ".

---

### Official Review · Reviewer_Rjdk · 2024-10-29

**Soundness:** 2
**Presentation:** 1
**Contribution:** 1
**Rating:** 3
**Confidence:** 3

**Summary:**

This paper tackles the constrained optimal consensus problem in a distributed directed graph setting. In particular, the paper tackles two constraint types, namely set constraints and general ones, where both scenarios have dedicated algorithms and convergence analysis provided in the paper. Numerical results are provided to strengthen the claims of the paper and the proposed algorithm is compared to an existing method.

**Strengths:**

In general, I appreciated the efforts made by the authors to provide a rigorous analysis of the algorithms discussed in the paper.

-	The paper emphasizes mathematical results by providing assumptions, definitions, and lemmas to then provide a convergence analysis of the method(s) they propose.
-	Certain claims are validated in the experiments, such as the faster convergence of the proposed method for set constraints when compared to an existing algorithm.

**Weaknesses:**

**The major weakness of this paper comes from the presentation.** Overall, I find the paper difficult to follow, and difficult to understand the extent of its contribution, which I detail further below:

-	**Many mathematical results are given without context:** In many cases, the variables, assumptions, definitions, and equations are present without further elaboration or justification. For example, Assumption 2.1 states that the considered graph should be strongly connected and balanced, but it is not explained what is the motivation behind this assumption, nor the terms are clearly defined. Another example is the content of Section 2.2 and Section 2.4 which both consist of several mathematical statements and results placed one after the other without really linking them. A third example would be equation (18) and the sentence surrounding it, where it is not clear what information should the reader note about this statement.
-	**Proposed algorithms given without context:** Similar to the previous point, the two algorithms proposed in the paper are given without context, lacking a justification of the different terms in (5), (12), and (13). Without providing these, it is very difficult for the reader to understand the originality, novelty, and intuition behind the proposed methods. Every term of these equations should be explained and the novel parts should be emphasized. Currently, it is not clear what parts of the proposed algorithms are new and which ones are parts of existing methods. Additionally, some terms in the proposed algorithms are not defined, for example, $T$ and $\alpha$ are not mentioned before (5).
-	**The motivation behind the proposed approaches is limited:** In Section 1.1, potential alternative approaches existing in the literature are presented. For the set-constrained case, it is stated in the paper that:
“One drawback of utilizing diminishing step sizes is a slower convergence rate. Therefore, it is desired to design a projection based algorithm with constant step-sizes.”
As for the general constraints, the authors mention:
“Besides the difficulties of projection operation, the existence of nonlinear inequality constraints has introduced significant complexities into stability analysis…”
These two statements are, in my opinion, not very clear motivations for proposing alternative solutions. *For the former case, is a diminishing step-size that significant of a drawback, are there other studies on this? Additionally, what other drawbacks do the existing algorithms have, and are they also solved by the proposed approach? For the latter case, what type of complexities have been introduced, are they just more complex to analyze or does a stability analysis not exist for those algorithms?*
-	**The manuscript requires proofreading:** I will give a few examples. (i) The Laplacian $\mathcal{L}$ of a graph is mentioned and used to make mathematical statements, before defining it in the next paragraph, at the beginning of page 3. (ii) At the beginning of Section 3.1, $f_i(q)$ is mentioned where $q$ has not been defined and is probably meant to be $x$. (iii) The algorithm used for comparison in Section 5 is sometimes referred to as DPS, sometimes as DSP. (iv) In Section 5, it is written that “The DPS algorithm (6)…”, but equation (6) is an equation about the equilibrium analysis. (v) The legend of Figure 1.b refers to equation (24), which is never mentioned in the main text, and is a seemingly unrelated equation found in the appendix.

Additionally, I find that **the simulation section can also be developed further**. At least one more optimization problem can be studied. There also seems to be a mistake in the expression of $f_i(x_i)$ in Section 5, as it does not correspond to a quadratic function as mentioned in the text.

**Questions:**

I have provided a few questions in the “Weaknesses” part (highlighted in italic). On top of them, I have the following questions:

-	I saw that a potential missing reference is:
Liu Q, Yang S, Hong Y. Constrained consensus algorithms with fixed step size for distributed convex optimization over multiagent networks. IEEE Transactions on Automatic Control. 2017 Mar 10;62(8):4259-65.
Can the authors detail how the method they propose compares to this reference?

-	How is the analysis different for undirected graphs? In which part does the directed aspect play a role in the methods provided?
-	In Section 5, are the presented results obtained through multiple independent runs of the algorithms, for example using a Monte-Carlo method, or are the results based on a single run?

---

### Official Review · Reviewer_8Wic · 2024-11-05

**Soundness:** 2
**Presentation:** 2
**Contribution:** 2
**Rating:** 3
**Confidence:** 4

**Summary:**

This paper proposes two projection-based distributed constrained optimal consensus algorithms for multi-agent systems under a directed graph. The proposed algorithms only require agents to exchange their relative state, which helps preserve privacy.  Theoretical result in terms of convergence rate is provided along with experiments to show the linear convergence of the proposed algorithms.

**Strengths:**

This paper studies the distributed constrained optimal consensus problem of multiagent systems under a directed graph and utilizes constant learning rates to achieve faster convergence compared with conventional decaying learning rate methods. The proposed methods are supported by typical convergence analysis and some experimental results.

**Weaknesses:**

1. The paper is not well organized and is hard to follow.
2. The experiments are limited. For example, only one synthetic data is tested and the experiments only show the convergence properties of the related errors. The comparison with SOTA does not look convincing in several aspects. First, what is the DSP /DPS algorithm? The two names of the algorithm both appear in the paper while the full name is missing and it is not clear what is the difference of the DSP/DPS(?) algorithm and the proposed method except for the learning rate. Second, a number of optimization methods with constraints have been reviewed in the Introduction part, however, it is not clear why they are not used as benchmarks in the experiments.
3. The writing needs improvement. Many equations have space issues and missing some sentences missing punctuation. For example, Line 403 "Stability Analysis: In this part, we have the following assumption" should be "Stability Analysis: In this part, we have the following assumption.", line 270 "equation (7)" should be "Equation (7)". Some symbols are not defined such as $T$ and $\alpha_i$ when they are used in the equations.

**Questions:**

See the weakness part.

---

### Official Review · Reviewer_YjWw · 2024-11-06

**Soundness:** 2
**Presentation:** 3
**Contribution:** 2
**Rating:** 5
**Confidence:** 4

**Summary:**

In this present paper, the authors presented the projection-based distributed constrained optimal consensus algorithms to solve distributed constrained optimal consensus problem of multi-agent systems. Under certain assumptions, they proved the convergence of these algorithms. Numerical examples were presented to illustrate the efficiency of the algorithms.

**Strengths:**

The presentation of the paper is well organized and the theoretical results are clear. The mathematics, as far as I have checked, are correct.

**Weaknesses:**

Beside the theoretical results, the efficiency of the proposed algorithms were not well exposed. For instance, my main concern is that it lacks of detailed comparison with existing works. The numerical examples are insufficient. There was only one baseline algorithm to be compared with the DPS (should be DSP) algorithm (6).  A number of works, including the reference the present paper, should be taken into considerations.

 As for the theoretical part, the technical proof is conventional. If the contribution of the paper was theoretical, there was lack of validation of its theoretical contribution, for instance, to be employed to analyze other DSP algorithms.

The paper needs thorough proofreading. There are a number of typos and grammatic errors. I can point out only a few them:

line 29: "construct“ should be "constructing";
line 45: "uses" should be "use"; ”has“ should be "have";
line 111: the second  word "graph" should be removed.
line 137: "spinning" should be "spanning".
line 484, "DPS" should be "DSP".
etc.

**Questions:**

1. How do the projection operators work for the optimization performance?  I suggested some ablation experiments to show the performance of these projection operators.
2. How to realize the projection under complicated constraints? For some complication constraints, it is always difficult to realize a proper projection from the outside to the constrained region. For instance, the variable regions of the constraints is nonconvex or the dimensions of the variables is very high.
3. How is the sliding mode approach used in the algorithm? I cannot see the details of the sliding mode in the present paper. So it is difficult to me to validate its novelty or contribution.

---

### Meta-Review · Area_Chair_jHv1 · 2024-12-20

**Metareview:**

This paper investigates the distributed constrained optimal consensus problem in multi-agent systems operating under directed graphs and introduces two projection-based algorithms to address this challenge. One algorithm handles set constraints, while the other is designed for more general constraints. Both methods rely solely on the exchange of relative state information among agents, enhancing privacy.

For set constraints, the authors transform the distributed optimization problem into a constrained leaderless consensus problem using a sliding mode approach. Building on this transformation, they extend their approach to develop a projection-based algorithm for general constraints. The proposed methods are proven to achieve an ergodic convergence rate in terms of first-order optimality residuals, ensuring effective and efficient solutions.

Theoretical analysis establishes the algorithms' convergence under specific assumptions, and numerical simulations validate their effectiveness. Results demonstrate the linear convergence of the proposed methods, illustrating their robustness and applicability in distributed optimization tasks for multi-agent systems. These algorithms offer a practical solution for achieving constrained optimal consensus while preserving privacy and computational efficiency.

At least three reviewers commented that the paper is not well written and the content of the paper has a difficult flow. Besides that, there are a number of technical issues raised about the efficiency of the algorithm, the sufficiency of the empirical results, and some of the assumptions. However, overall, the quality of presentation overshadows all other concerns.

**Additional Comments On Reviewer Discussion:**

The authors did not participate in the rebuttal process.

---

### Decision · Program_Chairs · 2025-01-22

Reject